# Distinct genetic architectures and environmental factors associate with host response to the γ2-herpesvirus infections

Neneh Sallah [1,2,14✉], Wendell Miley[3], Nazzarena Labo[3], Tommy Carstensen[1,4], Segun Fatumo[1,2,5], Deepti Gurdasani[1,6], Martin O. Pollard [1,4], Alexander T. Dilthey [7], Alexander J. Mentzer [8,9], Vickie Marshall[3], Elena M. Cornejo Castro [3], Cristina Pomilla[1,4], Elizabeth H. Young[1,4], Gershim Asiki [10], Martin L. Hibberd [2], Manjinder Sandhu [4], Paul Kellam [11,12], Robert Newton[5,16], Denise Whitby[3,16] & Inês Barroso [1,13,15,16✉]

Kaposi's sarcoma-associated herpesvirus (KSHV) and Epstein-Barr Virus (EBV) establish life-long infections and are associated with malignancies. Striking geographic variation in incidence and the fact that virus alone is insufficient to cause disease, suggests other co-factors are involved. Here we present epidemiological analysis and genome-wide association study (GWAS) in 4365 individuals from an African population cohort, to assess the influence of host genetic and non-genetic factors on virus antibody responses. EBV/KSHV co-infection (OR = 5.71(1.58–7.12)), HIV positivity (OR = 2.22(1.32–3.73)) and living in a more rural area (OR = 1.38(1.01–1.89)) are strongly associated with immunogenicity. GWAS reveals associations with KSHV antibody response in the *HLA-B/C* region ($p = 6.64 \times 10^{-09}$). For EBV, associations are identified for VCA (rs71542439, $p = 1.15 \times 10^{-12}$). Human leucocyte antigen (HLA) and trans-ancestry fine-mapping substantiate that distinct variants in *HLA-DQA1* ($p = 5.24 \times 10^{-44}$) are driving associations for EBNA-1 in Africa. This study highlights complex interactions between KSHV and EBV, in addition to distinct genetic architectures resulting in important differences in pathogenesis and transmission.

[1] The Wellcome Sanger Institute, Wellcome Genome Campus, Cambridge, UK. [2] London School of Hygiene & Tropical Medicine, London, UK. [3] Viral Oncology Section, AIDS and Cancer Virus Program, Frederick National Laboratory for Cancer Research, Leidos Biomedical Research Inc., Frederick, MD, USA. [4] Department of Medicine, University of Cambridge, Cambridge, UK. [5] MRC/UVRI at the London School of Hygiene & Tropical Medicine, Entebbe, Uganda. [6] Queen Mary University London, London, UK. [7] Institute of Medical Microbiology and Hospital Hygiene, Heinrich Heine University Düsseldorf, 40225 Düsseldorf, Germany. [8] Wellcome Centre for Human Genetics, University of Oxford, Oxford, UK. [9] Big Data Institute, Li Ka Shing Centre for Health Information and Discovery, University of Oxford, Oxford, UK. [10] African Population and Health Research Center, Nairobi, Kenya. [11] Department of Infectious Diseases, Imperial College London, London, UK. [12] Kymab Ltd, Babraham Research Complex, Cambridge, UK. [13] MRC Epidemiology Unit, University of Cambridge, Cambridge, UK. [14] Present address: London School of Hygiene & Tropical Medicine, London, UK. [15] Present address: Exeter Centre of ExCEllence in Diabetes (ExCEED), University of Exeter Medical School, Exeter, UK. [16] These authors contributed equally: Robert Newton, Denise Whitby, Inês Barroso. ✉email: Neneh.Sallah@lshtm.ac.uk; ines.barroso@exeter.ac.uk

The gamma-herpesviruses, Epstein−Barr Virus (EBV) and Kaposi sarcoma-associated herpesvirus (KSHV) are oncogenic viruses associated with several malignancies, including >95% of Burkitt's Lymphoma (BL) in parts of sub-Saharan Africa and 100% of cases of Kaposi's Sarcoma (KS), respectively[1,2]. While EBV is ubiquitous globally, the prevalence of, and deaths caused by several associated malignancies, display extensive geographic variation. In contrast, KSHV displays striking geographic variation that parallels associated KS incidence, with highest prevalence reported in sub-Saharan Africa[1,3]. The inter and intra-continental heterogeneity in the prevalence of KSHV and incidence of associated malignancies, while attributable to a number of factors including differences in pathogen prevalence, may also be influenced by many environmental determinants, host genetics and gene−environment interactions in different populations. Synergistic and antagonistic interactions have been reported for KSHV−EBV co-infection; B-lymphocytes act as a reservoir of latent infection, in dually infected cells, both viruses subvert the host immune response and inhibit lytic reactivation of each other[4–8]. While viral genetic factors have been extensively studied[9,10], host genetic factors and their interactions with the environment, leading to potential disease outcomes, are largely unknown and require investigation, particularly in Africa.

We and others have reported associations in the HLA class II region with EBV EBNA-1 antibody response[11–14]. On the other hand, host genetic studies for KSHV are lacking, with no genome-wide association study (GWAS) performed for traits associated with infection or diseases. The strong correlation of serological status between siblings (not explained by known risk factors), and the familial clustering of KSHV disease[15–19] are all highly suggestive of host genetic influence in addition to shared environmental risk factors. Candidate gene, whole-exome and next-generation sequencing methods have suggested Mendelian causes of rare paediatric KS cases as a result of inborn errors in immunity. However, the role of genetic variants in adult KS and in the viral control in asymptomatic individuals is less clear[20]. To date, most studies provided marginal evidence using statistically lenient $p$ value thresholds of between 0.01 and 0.05, which failed to correct for multiple testing to declare "association" for variants in immunomodulatory genes with virus biological function, pathogenesis and tumourigenesis[21–24]. In addition, they had other limitations including, very small sample sizes (between 1 and 350 cases), failure to adjust for environmental factors such as co-infection with other pathogens, or confounding by strong *HLA* associations with HIV and AIDS, and did not stratify controls by KSHV serostatus or include KSHV seronegative controls for comparison, and all lack replication in independent samples. Lastly, only two studies[23,25] have been conducted in African populations.

To overcome the limitations of previous studies and attempt to identify convincing associations with KSHV and EBV immune response traits, we assess systematic differences in >4000 individuals from an African population cohort, where both viruses are endemic, using socio-demographic and clinical data to assess intrinsic and environmental determinants of infection and then perform a GWAS using antibody responses as markers of infection. We use whole-genome sequence data, dense genotyping array data and imputation to a panel with African sequence data to identify genetic loci associated with both infections and attempt to replicate previously identified genetic loci in the context of the environment.

## Results

**Characteristics of samples in the Uganda General Population Cohort (GPC).** To investigate the seroprevalence of infections, we tested serum samples from 4365 individuals in the General Population Cohort (GPC) collected during medical survey round 22 (in 2011). The GPC is a population-based cohort in Kyamulibwa, rural south-west Uganda, comprising inhabitants of 25 neighbouring villages[26]. Participants were over the age of 13 years and belonged mainly (>70%) to the Baganda ethnolinguistic group. Villages were categorised according to urbanicity quartiles reflecting shared 'urban' characteristics based on differences in economic activity, civil infrastructure, and availability of educational and healthcare services as previously described[27], with 28% living in quartile 1 (very rural i.e. no educational facilities and no households with electricity) (Table 1). In this study, 91% of individuals were categorised as seropositive for EBV based on detectable IgG levels against either EBNA-1 or VCA[28] and 91% categorised as seropositive for KSHV based on detectable IgG levels against either ORF73, K10.5 or K8.1[29,30] (Table 1). HIV infection seroprevalence in this study was 6.5%, Hepatitis C virus (HCV) seroprevalence was 3.7% and Hepatitis B virus (HBV) infection had the lowest seroprevalence among the pathogens examined at 2.9% (Table 1). Nearly 95% of participants, 4134 individuals, were infected with at least one of these viruses. The majority of participants, 2743 (63%) were seropositive to at least two of the viruses tested, 23 (0.5%) participants were seropositive for four viruses and 231 (5.3%) participants were seronegative for all viruses (Supplementary Fig. 1). Co-infection with KSHV and EBV was the most common with >95% of dually infected individuals (Supplementary Table 1.). Co-infections of other pathogens with EBV or KSHV was similarly frequent and mirrored the seroprevalence estimates seen in the cohort (Supplementary Table 1).

**Table 1 Characteristics of individuals in the GPC.**

| Characteristic | $N = 4365$ | (%) |
|---|---|---|
| Sex | | |
| Female | 2537 | 58.1 |
| Male | 1828 | 41.9 |
| Age group (year) | | |
| 13−18 | 1078 | 24.7 |
| 19−39 | 1828 | 41.9 |
| 40+ | 1459 | 33.4 |
| Ethnic group | | |
| Baganda | 3328 | 76.2 |
| Other | 1037 | 23.8 |
| Education level attained | | |
| Primary | 1538 | 35.2 |
| Junior secondary | 1460 | 33.4 |
| Senior secondary | 905 | 20.7 |
| Tertiary | 223 | 5.1 |
| Urbanicity quartile | | |
| 1 | 1222 | 28.0 |
| 2 | 982 | 22.5 |
| 3 | 953 | 21.8 |
| 4 | 963 | 22.1 |
| KSHV | | |
| Positive | 3988 | 91.3 |
| EBV | | |
| Positive | 3956 | 90.6 |
| HIV | | |
| Positive | 284 | 6.5 |
| Hepatitis B | | |
| Positive | 126 | 2.9 |
| Hepatitis C | | |
| Positive | 161 | 3.7 |

*N* the number of unique individuals.

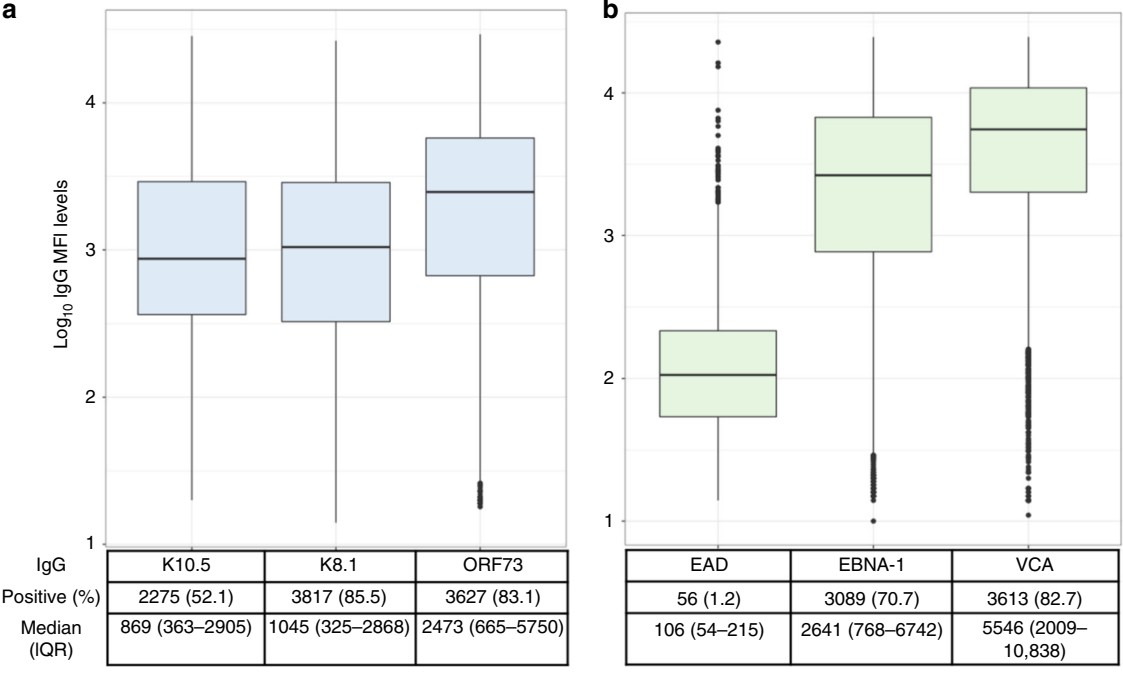

**Fig. 1 Inter-individual variability in IgG antibody responses to KSHV and EBV ($n = 4365$ individuals). a** Distribution of anti-KSHV antibody levels ($\text{Log}_{10}$ MFI) for K10.5 (min = 0, max = 28,514), K8.1 (min = 0, max = 26,468) and ORF73 (min = 0, max = 29,308). **b** Distribution of anti-EBV antibody levels ($\text{Log}_{10}$ MFI) for EAD (min = 0, max = 22,747), EBNA-1 (min = 0, max = 24,696), VCA (min = 11, max = 24,754). Seropositivity to all antigens and median (centre) plus IQR displayed in the text box.

To investigate the inter-individual variation of IgG antibody responses marking different stages of the viral life cycles (latent vs lytic) to KSHV and EBV infections, distributions of antibody levels were determined to the latent (ORF73, K10.5, EBNA-1) and lytic antigens (K8.1, VCA and EAD) in 4365 individuals. All antibody responses were highly variable across individuals as shown by the wide range of median fluorescent intensity (MFI) (Fig. 1a). For KSHV, seropositivity to the K8.1 and ORF73 were both high >80% with similar distributions observed for all anti-KSHV IgG (Fig. 1a). For EBV, seropositivity to EBNA-1 was highest at 83% (Fig. 1b) with the lowest antibody responses observed for EAD (Fig. 1b). The IgG responses were modestly phenotypically correlated ($r^2$: 0.1−0.43) (Supplementary Fig. 2).

**Environmental determinants of KSHV IgG antibody response**. To assess the contribution of environmental determinants on KSHV serostatus (i.e. seropositive vs seronegative) and specific IgG antibody response levels [MFI], we investigated 4365 individuals from the GPC[26]. We selected age, sex, socio-demographic and infection status factors (Table 1) that could potentially influence serostatus or the levels of seroreactivity to KSHV antigens. Following a multivariate logistic regression analysis, being EBV positive had the largest effect on KSHV serostatus (OR = 5.68, 95% CI = 4.55−7.06), and being male (OR = 1.35, 95% CI = 1.17−1.57) was also significantly associated with seropositivity (Fig. 2a). Living in a lower urbanicity quartile, lower educational attainment and HIV negativity were also significantly associated with seropositivity in individuals albeit with modest effect sizes (Fig. 2a and Supplementary Data 1). We further assessed the contribution of the same factors on IgG antibody levels to all antigens using a multivariate linear regression analysis. Similar to serostatus, anti-KSHV IgG levels were significantly higher in EBV seropositive individuals (OR = 2.19, 95% CI = 1.88−2.22) and higher in males than in females, particularly, for anti-ORF73 (OR = 1.17, 95% CI = 0.08−0.148) (Fig. 2b and Supplementary Data 1). HIV seropositivity was significantly

associated with decreased IgG antibody levels (OR = 0.86, 95% CI = 0.73−0.93), as were higher education levels, particularly tertiary education attainment (OR = 0.84, 95% CI = 0.72−0.92) (Fig. 2b). Age, ethnolinguistic group, HBV and HCV co-infections had no significant associations with KSHV serostatus or antibody levels. Full association results can be found in Supplementary Data 1.

**Environmental determinants of EBV IgG antibody response**. We then assessed the contribution of environmental determinants (Table 1) on EBV serostatus (i.e. seropositive vs seronegative) and the levels of EBV antigen-specific IgG antibodies. KSHV seropositivity (OR = 5.71, 95% CI = 4.58−7.12) had the largest effect on EBV serostatus (Fig. 3a). Living in an urbanicity quartile of 3 (OR = 1.38, 95% CI = 1.01−1.89) and HIV infection (OR = 2.22, 95% CI = 1.32−3.73) were also significantly associated with seropositivity to EBV (Fig. 3a and Supplementary Data 2). Unlike KSHV, sex, or education level were not significantly associated with EBV serostatus (Supplementary Data 2). KSHV seropositivity was also significantly associated with lower IgG antibody levels for all EBV antigens (Fig. 3b and Supplementary Data 2). HIV seropositivity was associated with higher antibody levels to lytic antigens VCA and EAD (OR = 1.37, 95% CI = 1.27−1.47) but not the latent antigen EBNA-1. Living in a higher urbanicity quartile of 3 was also associated with increased antibody levels to EBNA-1 (OR = 1.11, 95% CI = 1.05−1.18) but not the lytic antigens (Fig. 3b and Supplementary Data 2). Anti-EAD was the only antibody that decreased with living in an urbanicity quartile of 4 (OR = 0.88, 95% CI = 0.85−0.91) (Fig. 3b and Supplementary Data 2). Ethnicity, education level and HCV co-infection had no significant influence on IgG levels (Supplementary Data 2).

**Genetic determinants of anti-KSHV and anti-EBV antibody responses**. To assess the contribution of human genetic variation

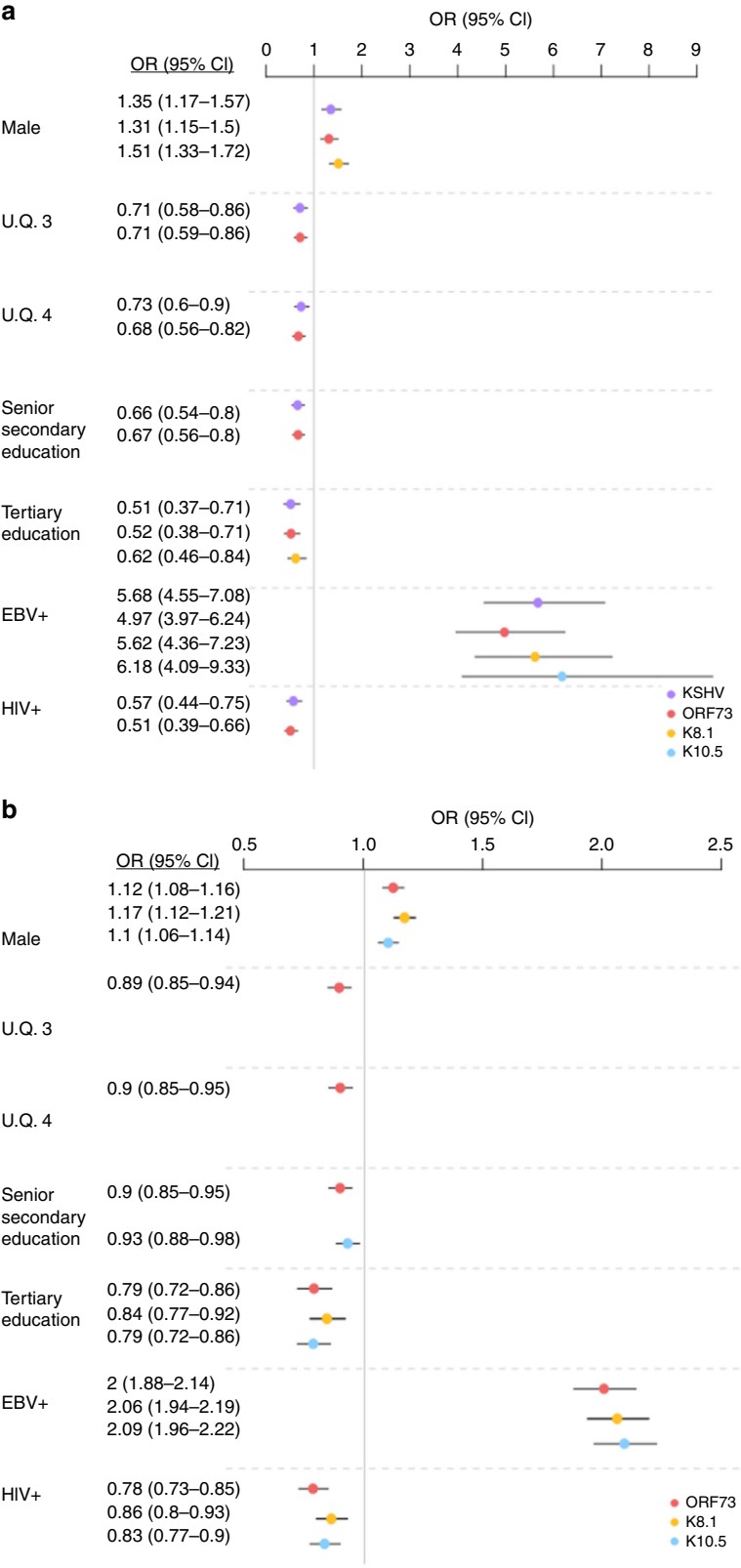

**Fig. 2 Predictors of KSHV IgG antibody response.** Odds ratios (ORs) and 95% confidence intervals (CI) of significant associations ($p < 0.006$) between factors and IgG variable as determined based on serologies in the 4365 individuals from the GPC. **a** Predictors of IgG serostatus. ORs and 95% CI were estimated following a multivariate logistic regression analysis with serostatus (seropositive vs seronegative) as response variable. **b** Predictors of IgG response level. ORs and 95% CI were estimated following a multivariate linear regression analysis with $\log_{10}$-transformed MFI as response variable. UQ urbanicity quartile. Dots represent the ORs and are coloured by IgG variable: KSHV (seropositivity to any antigen)—purple, ORF73—red, K8.1—yellow, K10.5—light blue; lines represent the 95% confidence intervals. Full results from this analysis are provided in Supplementary Data 1.

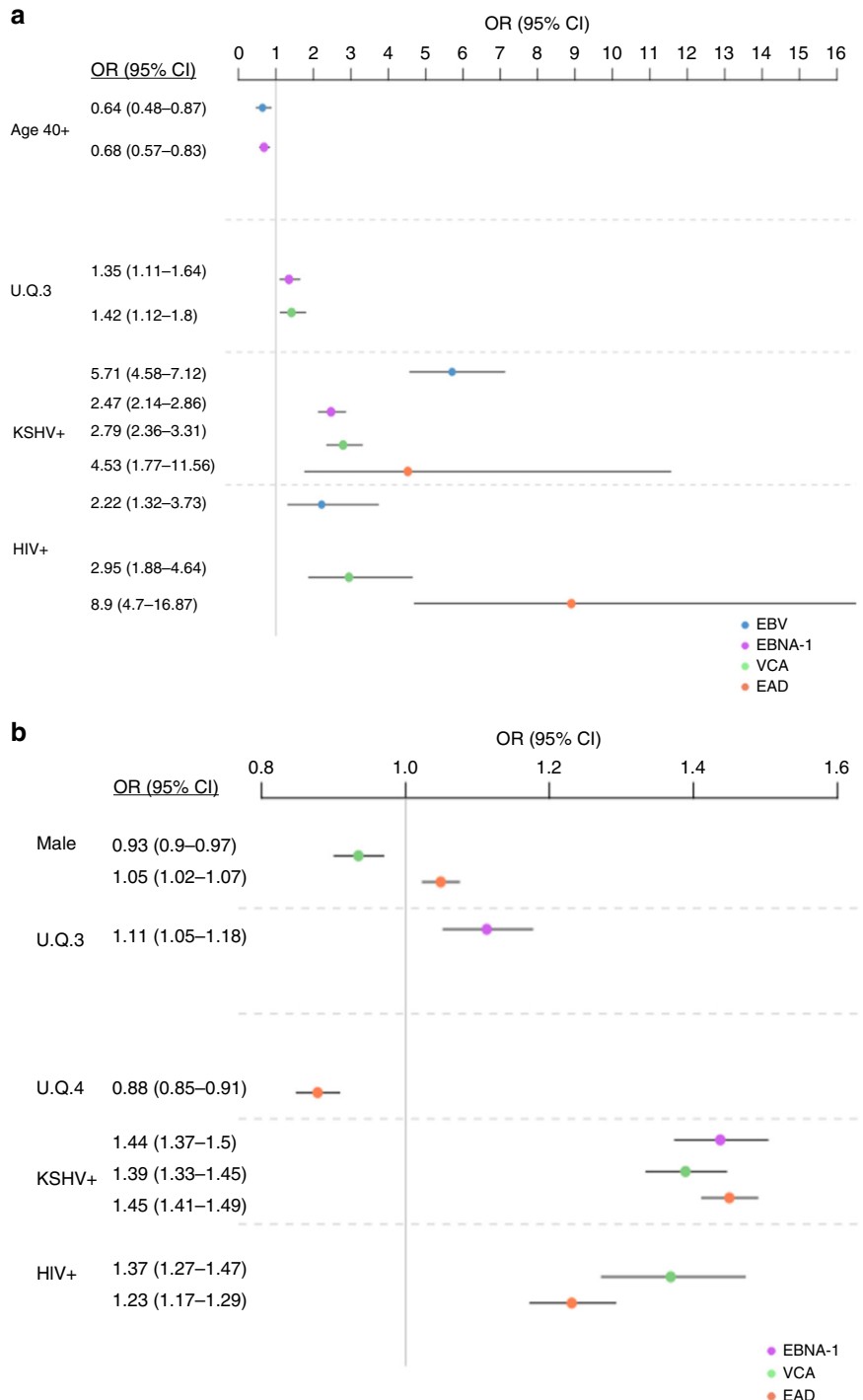

**Fig. 3 Predictors of EBV IgG antibody response.** Odds ratios (ORs) and 95% confidence intervals (CI) of significant associations ($p < 0.006$) between factors and IgG variable as determined based on serologies in the 4365 individuals from the GPC. **a** Predictors of IgG serostatus. ORs and 95% CI were estimated following a multivariable logistic regression analysis with serostatus (seropositive vs seronegative) as response variable. **b** Predictors of IgG response level. ORs and 95% CI were estimated following a multivariable linear regression analysis with $\log_{10}$-transformed MFI levels as response variable. UQ urbanicity quartile. Dots represent the ORs and are coloured by IgG variable (EBV (seropositivity to EBNA-1 or VCA)—blue, EBNA-1—magenta, VCA—light green, EAD—orange), lines represent the 95% confidence intervals. Full results from this analysis are provided in Supplementary Data 2.

on antibody responses to KSHV ($N = 4365$) and EBV ($N = 4365$), we first explored the heritability of anti-KSHV and anti-EBV IgG serological traits in the GPC. As most individuals in the cohort are seropositive (>90%) presumably from childhood[30], those who present as seronegative are also likely to have been infected and thus, produced antibodies at very low/undetectable levels; as such, they are also informative and useful to include in

analysis. In this study, heritability estimates were high for anti-ORF73 ($h2 = 33\%$, 95% CI = 24–42%), anti-K8.1 ($h2 = 25\%$, CI = 16–35%) and anti-K10.5 ($h2 = 32\%$, 95% CI = 23–41%), and for anti-EBNA-1 ($h2 = 36\%$, 95% CI = 28–45%) and anti-VCA IgG ($h2 = 32\%$, 95% CI = 23–41%). After correction for unmeasured shared environmental correlation, accounted for by GPS coordinates[11,31], our estimates were reduced yet still

**Table 2 Lead association signals with anti-KSHV IgG antibody levels.**

| Trait | Chr:Pos(b37) | Variant | Mapped gene(s)[a] | Consequence[a] | EA | EAF (%) | $p$ | $\beta$ (95% CI) |
|---|---|---|---|---|---|---|---|---|
| ORF73 | 5:37832167 | rs114429578 | *GDNF* | Intergenic | T | 3.1 | $4.60 \times 10^{-08}$ | 0.36 (0.23–0.49) |
| | 6:32584693 | rs510205 | *HLA-DQA1* | Intergenic | G | 9.4 | $5.53 \times 10^{-08}$ | 0.21 (0.14–0.29) |
| K10.5 | 6:31262619 | rs111664408 | *HLA-B/ HLA-C* | Intergenic | G | 10.6 | $6.64 \times 10^{-09}$ | −0.22 (−0.29 to −0.15) |
| | 10:124833806 | rs10794590 | *ACADSB* | Intergenic | T | 25.7 | $4.59 \times 10^{-08}$ | −0.14 (−0.19 to −0.09) |

*EA* effect allele, *EAF* effect allele frequency.
[a]Annotated using Ensembl VEP.

significant for all traits as previously found in our pilot study[11], anti-ORF73 ($h^2_{Adjusted} = 20\%$, 95% CI = 13−26%), anti-K8.1 ($h^2_{Adjusted} = 13\%$, 95% CI = 7–19%) and anti-K10.5 ($h^2_{Adjusted} = 18\%$, 95% CI = 11–24%), and for anti-EBNA-1 ($h^2_{Adjusted} = 20\%$, 95% CI = 14–27%) and anti-VCA IgG ($h^2_{Adjusted} = 18\%$, 95% CI = 13–25%).

To further investigate the genetic determinants of response to infection in this population, we conducted GWAS for each continuous antibody trait combining whole-genome sequencing and dense genotyping data with imputation to a merged 1000 Genomes phase 3, AGVP[32] and UG2G reference panel[33]. Stringent sample and variant QC left 4365 individuals with all phenotypes and ~17 M SNPs across autosomal markers and X-Chromosome for analyses (see 'Methods'). Homogeneity in the study population was previously ascertained by principal components analyses with AGVP populations as a reference panel[11]. This model accounted well for population structure and cryptic relatedness as shown by genomic inflation factor ($\lambda$) ~1.0 for all traits (Supplementary Fig. 3). Correcting for multiple testing, accounting for the lower linkage disequilibrium (LD) in African populations and using an FDR of 5%, the genome-wide significance threshold was adjusted to $p < 1 \times 10^{-8}$ [11,33]. Following GWAS of quantitative anti-ORF73, anti-K8.1 and anti-K10.5 IgG levels for 4365 individuals, one locus in association with K10.5, rs111664408-G ($p = 6.64 \times 10^{-09}$, $\beta$, 95% CI = −0.2,−0.29 to −0.15) on chromosome 6, mapped as intergenic between *HLA-B* and *HLA-C*, reached genome-wide significance threshold (Table 2 and Fig. 4a). In addition, the following variants reached the standard genome-wide significance threshold ($p < 5 \times 10^{-8}$) used in European studies which we highlight as they merit being prioritised for replication in other studies; for K10.5, rs10794590-T ($p = 4.59 \times 10^{-08}$, $\beta$, 95% CI = −0.14, −0.19 to −0.09), an intergenic variant on chromosome 10 (Table 2 and Fig. 4b) and for ORF73, rs114429578-T ($p = 4.60 \times 10^{-08}$, $\beta$, 95% CI = 0.36, 0.23−0.49) on chromosome 5 mapped in *GDNF* (Table 2 and Fig. 4c). Another plausible signal rs510205-G ($p = 5.53 \times 10^{-08}$, $\beta$, 95% CI = 0.21,0.14–0.29) on chromosome 6 mapped in *HLA-DQA1* (Table 2) was also associated with anti-ORF73 IgG. In the GTEx database, rs10794590 (associated with K10.5 IgG levels) was associated with the differential expression of *ACADSB* ($p = 2.6 \times 10^{-9}$) and *IKZF5* ($p = 1.5 \times 10^{-3}$) in oesophagus mucosa and colon tissues. We also found rs510205 (associated with ORF73 IgG) associated with the differential expression of the following eight genes, *HLA-DQA2, HLA-DQB2, HLA-DQB1, HLA-DRB1, HLA-DRB6, HLA-DQA1, CYP21A1P* and *LY6G5B* in whole blood ($p < 3.8 \times 10^{-6}$). Expression data for our other association signals were unavailable in the GTEx database. We were unable to refine signals linked to *HLA* alleles with no significant associations identified with ORF73 or K10.15 IgG levels (Supplementary Data 3). We also assessed whether variants within 23 genetic loci that been previously identified in other studies with marginal significance as associated with classic KS, KSHV seropositivity, KSHV viral load, antibody response or primary effusion lymphoma[21,23–25,34–36] were present in this

study and had plausible signals. In this study, 21 out of 31 variants were typed/imputed and were in *IL6, IL8RB, IL13, IL4, IL12A, FCγRIIIA, HLA* and *IRAK1*[21,23,24,34–36], data for KIR types[22] were unavailable in this study. Despite adequate power (80%), we did not detect significant associations ($p > 0.1$) of any of the previous suggested variants for any of the traits in this study (Supplementary Table 2).

For EBV infection, GWAS of 3289 individuals, replicated our previous finding[11] rs9272371 in *HLA-DQA1* ($p = 5.24 \times 10^{-33}$, $\beta = -0.37$) and identified a distinct secondary signal, rs3129867 in *HLA-DRA* ($p_{cond} = 6.01 \times 10^{-11}$, $\beta = -0.17$) (Supplementary Table 3). To maximise power, we combined our dataset with 1076 individuals from our pilot study[11] and performed GWAS for EBV response in 4365 individuals (Supplementary Fig. 5). We found stronger statistical associations for anti-EBNA-1 IgG antibody responses in the *HLA* class II region (Table 3 and Fig. 5). The C-allele at our lead SNP rs9272371 in *HLA-DQA1* ($p = 3.63 \times 10^{-44}$, $\beta = -0.37$) was associated with lower antibody levels (Fig. 5a and Table 3), with the presence of a distinct secondary signal rs3129867 in *HLA-DRA* (Fig. 5b and Table 3) with the G allele associated with lower antibody levels ($p_{cond} = 2.78 \times 10^{-11}$, $\beta = -0.17$). We also identified significant associations for anti-VCA IgG antibody levels also in the HLA-class II region (Fig. 5c and Supplementary Fig. 5). The C-allele of the lead SNP rs71542439, also in *HLA-DQA-1*, associated with lower antibody levels ($p = 1.15 \times 10^{-12}$, $\beta = -0.21$) (Table 3). To fine map the associations observed in the *HLA* region, we tested the presence or absence of 117 four-digit imputed classical HLA alleles. *HLA-DRB1*15:01* showed the lowest $p$ value for association with EBV EBNA-1 ($p = 4 \times 10^{-4}$) (Supplementary Data 4) and had very weak LD with other alleles ($r^2 < 0.02$) (Supplementary Data 5). We did not replicate *HLA-DRB1*0701* ($p = 0.86$) or *HLA-DQB1*0301* ($p = 0.84$) alleles (Supplementary Data 4) previously associated with EBNA-1 IgG in individuals of European descent[13,14]. We also did not identify any significant associations between any classical *HLA* alleles and VCA IgG levels (Supplementary Data 4).

**Trans-ancestry meta-analysis and fine mapping of anti-EBNA-1 IgG response.** To investigate the portability of findings across ancestry, we used MANTRA[37] to perform a genome-wide trans-ancestry meta-analysis for anti-EBNA IgG responses, with association summary statistics of 4365 individuals from our Ugandan GWAS combined with 2162 seropositive individuals from the 1000 Genomes-imputed European ancestry GWAS[13], giving a total of 6527 individuals with ~4.9 million shared SNPs for analysis. Using a threshold of $\log_{10}$ Bayes Factor (BF) > 6[38] we replicated strong evidence of association in the *HLA* class II region with lead SNP rs6927022 ($\log_{10}BF = 44.3$) (Supplementary Fig. 6) previously identified as the lead association SNP in the European ancestry study[13]. Our Ugandan lead SNP rs9272371 ($\log_{10}BF = 43.2$) displayed heterogeneity in effect sizes in the two studies ($p_Q = 3.56 \times 10^{-8}$) as identified in our pilot study. To

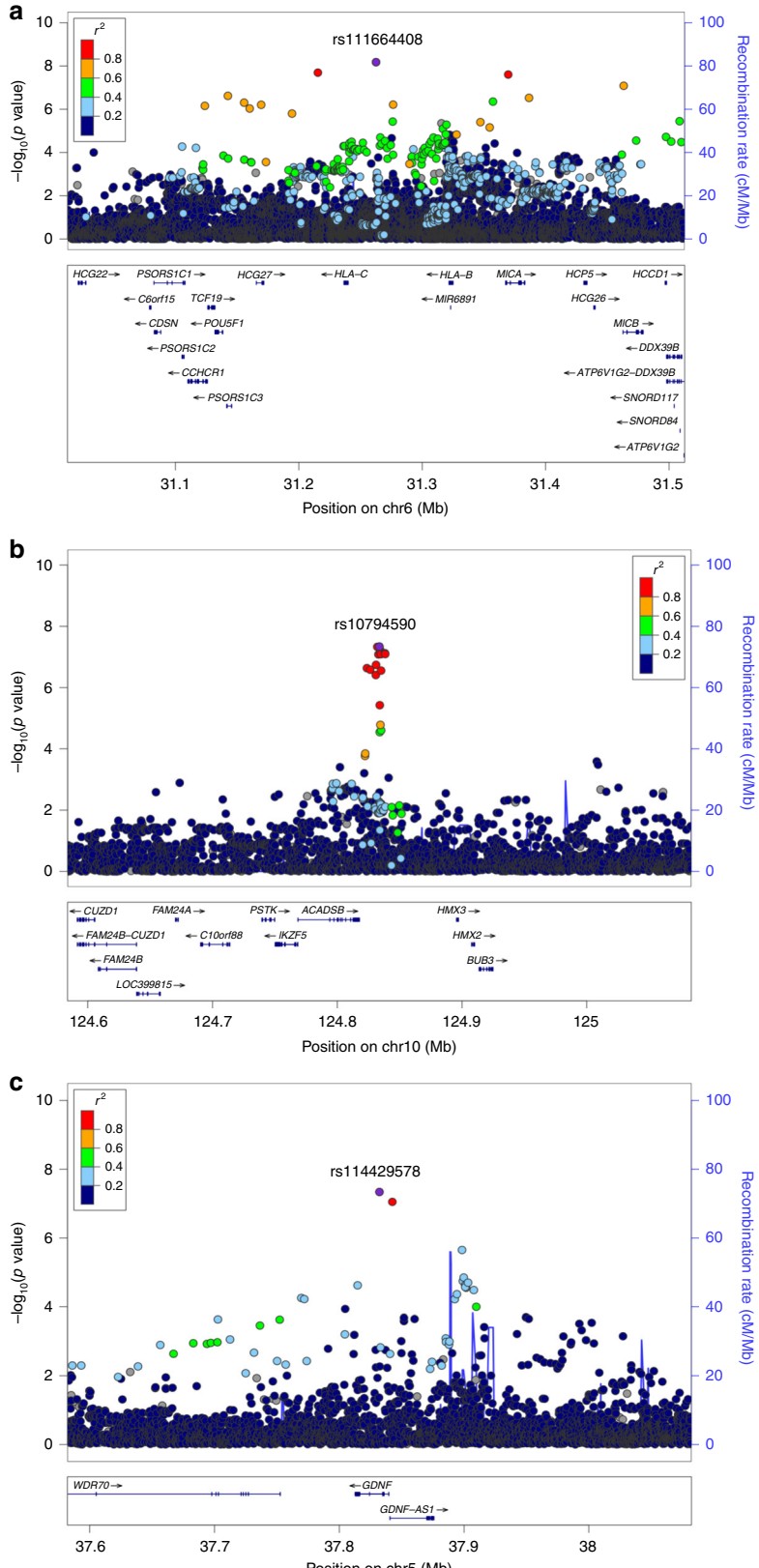

**Fig. 4 Regional association plots for lead SNPs associated with anti-KSHV IgG levels $p < 5 \times 10^{-8}$), $n = 4365$ individuals. a** Association on chromosome 6 in *HLA* class I region with anti-K10.5 IgG (rs111664408, $p = 6.64 \times 10^{-09}$). **b** Association on chromosome 10 in the *ACADSB* region with anti-K10.5 IgG (rs10794590, $p = 4.59 \times 10^{-08}$). **c** Association on chromosome 5 in *GDNF* region with anti-ORF73 IgG (rs114429578, $p = 4.60 \times 10^{-08}$). GWAS performed using linear mixed model accounting for kinship in GEMMA. The lead SNPs are labelled and coloured in purple. LD ($r^2$) was calculated based on Ugandan SNP genotypes.

**Table 3 Loci with significant evidence of association with anti-EBV IgG levels ($N = 4365$).**

| Trait | Chr:Pos(b37) | Variant | Nearest gene[a] | Consequence[a] | EA | EAF (%) | p | β (95% CI) |
|---|---|---|---|---|---|---|---|---|
| EBNA-1 | 6:32604654 | rs9272371 | HLA-DQA1 | Intron | C | 26.9 | $3.63 \times 10^{-44}$ | −0.37 (−0.43 to −0.31) |
| EBNA-1 | 6:32404220 | rs3129867 | HLA-DRA | Upstream | G | 49.2 | $2.10 \times 10^{-24*}$ | −0.25 (−0.30 to −0.20) |
| VCA | 6: 32630257 | rs71542439 | HLA-DQB1 | Intron | T | 21.1 | $1.15 \times 10^{-12}$ | −0.21 (−0.27 to −0.15) |

*EA* effect allele, *EAF* effect allele frequency.
*$p_{cond}$ on lead SNP, β (95% CI) = $2.78 \times 10^{-11}$, −0.15 (−0.18 to −0.11).
[a]Annotated using Ensembl VEP.

further, refine genetic association signals, using MANTRA results we generated 99% credible sets most likely to drive association signals and contain causal variants (or tagging unobserved causal variants) and compared fine-mapping intervals for each associated lead SNP by analysing the variants 500 kb up- and downstream of the lead SNP in the combined dataset as described previously[39,40]. This resulted in three distinct SNPs in the credible set, rs6927022, rs9272371 and rs9274247 (Supplementary Table 4) further confirming that rs6927022 does not fully drive associations in the Ugandan population. No other locus was found to be in association with anti-EBNA-1 IgG response.

## Discussion

In this study, we assessed the seroprevalence of KSHV and EBV infections along with the burden of co-infections and the influence of intrinsic, environmental and genetic factors on IgG antibody responses to infection in a rural African population cohort of >4000 individuals. Serological assays are useful in assisting diagnosis, and in improving our understanding of prevalence and transmission of infection, in addition to understanding virology and host immunity. The presence of antibodies representing host immune response are commonly used as diagnostic markers for stage of infection and thus, seroprevalence is widely used as a measure of the frequency of infections in a population[41]. In the GPC, prevalence estimates of KSHV and EBV are high (>90%) and consistent with previous findings in Uganda[42–46]. For KSHV, we found increased antibody seroprevalence and IgG levels in males compared to females as previously reported[30]. For EBV, while higher age (>40 years) was associated with modest decrease in antibody response (OR = 0.64), the background antigenic milieu in this population is different to western populations; it is likely that while antibody levels increase in childhood, they could decay with time (in adulthood) as a result of immunomodulation by other factors as observed in other studies[47] or as a result of exposure to other pathogens. Environmental factors such as co-infection with other pathogens have been studied and found to influence variation in seroprevalence estimates, and antibody responses associated with infection[45,48]. This is consistent with the results presented in this study, where we found HIV infection to be associated with IgG antibody response levels to KSHV and EBV infection. While HIV infection was associated with lower antibody responses to KSHV, the inverse was observed for EBV antibodies. Similarly, lower urbanicity and educational attainment were associated with a decrease in antibody levels to KSHV and increase to EBV (Figs. 2 and 3). This could be due to differential effect of shared epidemiological risk factors for HIV, EBV and KSHV, and/or as a result of the mutual inhibition of KSHV and EBV that has been observed in other studies[5,6]. Here, we also found that KSHV and EBV co-infections influence antibody responses to each other, highlighting the direct or indirect interaction between the two viruses, which has been previously observed for viral shedding among Ugandan children and their mothers and also in a Cameroon KS case control study[28,48]. As Uganda is also a malaria

endemic area and studies have reported co-infection with *P. falciparum* and other parasites as having an influence on antibody titre[45,46,49], it would be useful to also incorporate this in future study design. While the cohort is indeed rural, a marked variation in levels of urbanicity across the villages, largely attributable to differences in economic activity, civil infrastructure, and availability of educational and healthcare services has been previously described[27]. It is interesting that even within a rural setting different levels of 'urbanicity' are associated with antibodies supporting broader observations contrasting higher seroprevalence in rural vs urban settings[46,48,50].

In addition to environmental factors described above, host genetic factors (which might also be influenced by the environment) can also contribute to phenotypic variation in infectious disease response traits. To explore the proportion of variance in antibody responses attributable to the host genetics, we investigated the heritable component of the IgG response to KSHV and EBV antigens in individuals with genotype data, adjusting for unmeasured environmental correlation using spatial distances and similar methodology used in previous studies[51]. These analyses show that while KSHV and EBV antibody response traits were heritable there is overestimation in heritability prior to accounting for shared environment, which is consistent with our previous findings in Uganda[11]. The heritability estimates for KSHV ($h^2 = 13–19\%$) are also lower than the reported estimate of 37% in a Mexican-American ancestry cohort[52]; this deflation in estimates is similar to what we had previously observed for our pilot study of EBV in Uganda ($h^2_{EBNA-1} = 11\%$) compared to Mexican-Americans or Europeans ($h^2_{EBNA-1} = 37–43\%$)[11,12,53,54]. These differences in the GPC compared to other populations could be attributed to a number of factors including differences in study/assay design, locus or allelic heterogeneity influencing traits, not fully adjusting for environmental effects and differences in gene−environment interactions.

Conducting GWASs for KSHV-associated diseases is challenging as sample sizes are limiting to achieve power for reliable detection of loci. Prior to this study, GWAS had not been performed for any KSHV phenotype. As previous studies have shown correlation of IgG levels with the development of disease[55,56], here we used IgG antibody responses as a proxy for disease progression. While anti-KSHV IgG traits are partly heritable after accounting for shared environment, despite having >80% power to detect signals of moderate to large effect sizes ($β > 0.15$, for MAFs >25%) for all traits, only one genome-wide significant locus ($p < 1 \times 10^{-8}$) was identified, the variant rs111664408 in *HLA-B/HLA-C* ($p = 6.64 \times 10^{-09}$) on chromosome 6 contributing to inter-individual variability in K10.5 IgG responses. Nonetheless, using the standard genome-wide significance threshold ($p < 5 \times 10^{-8}$), two candidate loci associated with anti-ORF73 and anti-K10.5 IgG response levels were identified, reflecting the latent stage/history of infection. For anti-ORF73 IgG, relatively strong associations were in *GDNF* on chromosome 5 (rs114429578, $p = 4.60 \times 10^{-08}$) and in *HLA-DQA1* (rs510205, $p = 5.53 \times 10^{-08}$) (Table 2 and Fig. 4). While

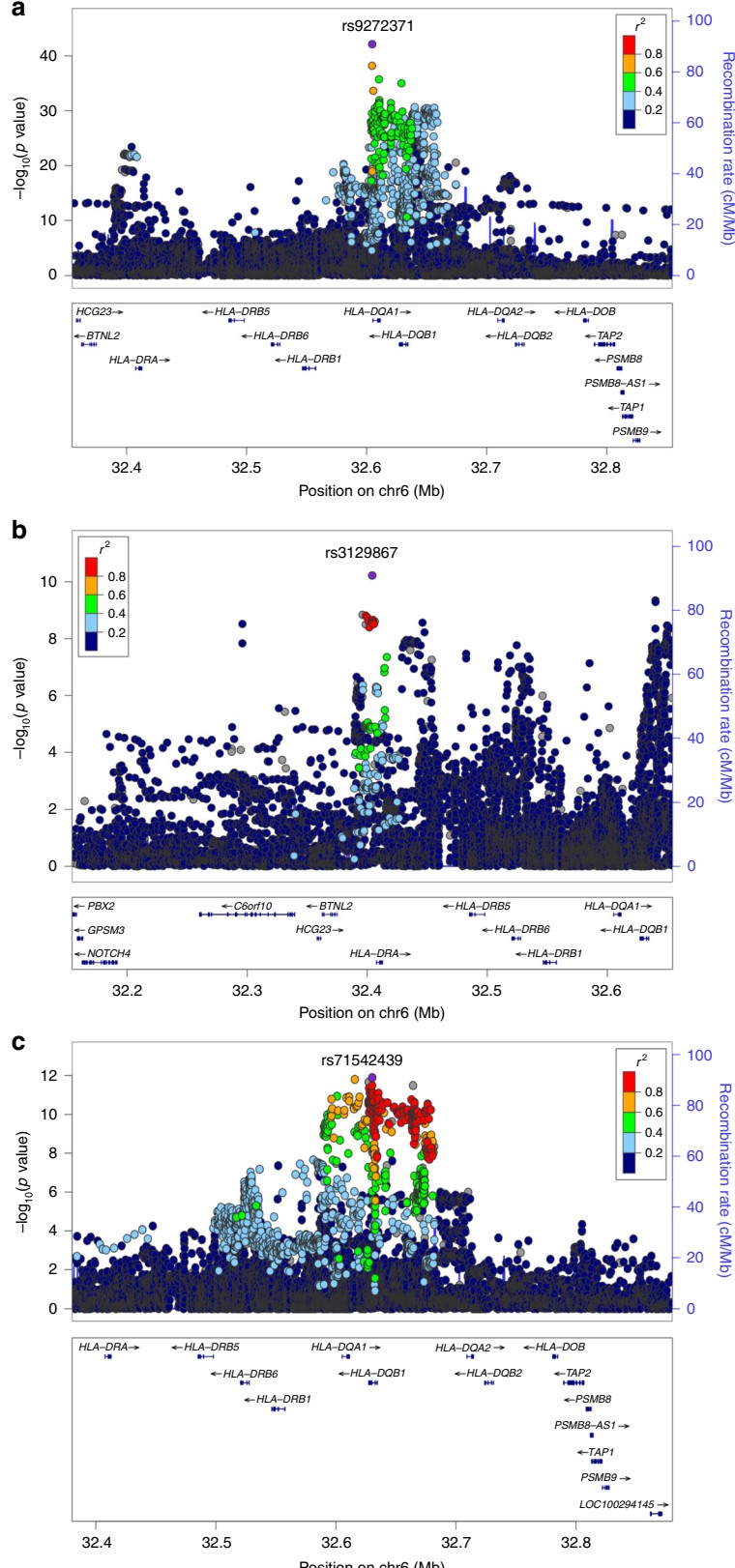

**Fig. 5 Regional association plots for lead SNPs associated with anti-EBV IgG levels (p < 1 × 10⁻⁸), _n_ = 4365 individuals. a** Lead association on chromosome 6 in _HLA-DQA1_ with anti-EBNA-1 IgG (rs9272371, $p = 3.60 \times 10^{-44}$). **b** Secondary association on chromosome 6 in HLA-DRA with anti-EBNA-1 IgG (rs3129867, $p_{cond} = 2.78 \times 10^{-11}$). **c** Lead association on chromosome 6 in the _HLA-DQB1_ with anti-VCA IgG (rs71542439, $p = 1.15 \times 10^{-12}$). GWAS performed using linear mixed model accounting for kinship in GEMMA. The lead SNPs are labelled and coloured in purple. LD ($r^2$) was calculated based on SNP genotypes in PLINK.

*GDNF* (encoding glial cell line-derived neurotrophic factor) has not been previously implicated in KSHV pathogenesis, it has been reported to be critical in the maintenance of latency for Herpes-simplex viruses infections[57]. The HLA class I and II regions, however, have been reported to play a role in the pathogenesis of KSHV and KS; nevertheless, previous genetic association studies have failed to identify convincing and reproducible associations at these loci[23,25]. Activation of CD4+ T cells is particularly important for anti-KSHV immunity and in vitro studies have shown the CD4+ T cells can inhibit viral replication in KSHV-infected tonsillar B cells[58,59]. Like EBV, KSHV has evolved strategies to evade immune detection, including negatively regulating the process by which HLA class II molecules present antigens to CD4+ T cells, thereby promoting its survival. Recently, ORF73 and v-IRF3 (K10.5) have been reported to inhibit MHC class II peptide presentation by blocking the transcription of the class II transactivator (CIITA), a master regulator of class II expression[60–62]. These findings suggest that variation in genes that play a role in modulating the immune response, particularly, T-cell immunity could contribute to inter-individual differences in anti-ORF73 and anti-K10.5 IgG levels to control KSHV viral infection, and thus, merit prioritisation in further replication studies in other populations.

For EBV infection, we and others have previously confirmed the presence of variants in the *HLA* Class II region driving associations for EBV EBNA-1 antibody response in different populations[11–14]; here we replicated these findings in *HLA-DQA1* (rs9272371, $p = 3.63 \times 10^{-44}$) (Fig. 5a) and identified an additional distinct variant (rs3129867, $p_{cond} = 2.78 \times 10^{-11}$) in *HLA-DRA* (Fig. 5b) contributing to IgG levels in Uganda that was not identified as distinct in our initial pilot study[11]. We also identified an association at rs28394498 ($p = 1.15 \times 10^{-12}$) in *HLA-DQA1* with anti-VCA IgG levels (Fig. 5c). We attempted to fine map the *HLA* region; *HLA-DRB1\*1501* present in 1.1% of individuals in this study was the most significant association signal ($p = 4.7 \times 10^{-4}$, $\beta = 0.42$) for EBNA-1 IgG. Interestingly, however, it did not explain our lead GWAS signal in *HLA-DQA1*. We also did not replicate *HLA* alleles *DRB1\*0701* and *DQB1\*0301* previously associated with EBNA-1 IgG levels in two GWASs consisting individuals of European descent[13,14]. Both studies, however, retained the significance of their lead GWAS SNPs in the *HLA* region after adjusting for the effect of the lead HLA alleles (residual $p < 2.6 \times 10^{-11}$)[13,14], and thus, could not fully explain the lead GWAS signal, which is consistent with our findings. This suggests the relevance of additional alleles or a non-HLA effect that did not reach significance or is absent in our study. It is also likely that allelic/haplotypic heterogeneity in Africans vs Europeans, an expression modulating effect or HLA imputation errors play a role particularly, given the limited representation of Africans in the multi-ethnic reference panel[63]. Trans-ancestry meta-analysis of European and African individuals in the *HLA* class II region confirmed distinct association signals in the two populations. As with our previous findings, rs9272371 and rs6927022 are likely to be distinct variants in the *HLA* class II region, with a single signal in Europeans (rs6927022) and a signal mostly driven by rs9272371 in Uganda (Supplementary Table 4). Thus, future studies should aim to further refine signals in larger sample sizes in individuals of African descent and using HLA panels based on African data which are currently in development.

In summary, we characterise systematic differences in >4000 individuals with evidence of infection by KSHV and EBV in Uganda and identify determinants of response to infection. The availability of data on environmental covariates such as co-infection with other pathogens allowed us to capture genetic variation independently of the environment. Co-infection was indeed a strong predictor of inter-individual variability in antibody responses and highlighted significant and complex interactions between EBV and KSHV. Despite similar heritability and power for KSHV and EBV traits, it was interesting to observe less genome-wide statistically significant associations for the former; this suggests differences in underlying genetic architectures between the traits. It is possible that variants with very small effect sizes or low-frequency variants (i.e. EAF <0.5%) influence inter-individual variability in KSHV immune responses; however, this study is under-powered to detect low-frequency variants with small effect sizes and neither the genetic data nor the tools used here are optimal for accurate rare variant detection. As sub-Saharan Africa sustains such high transmission compared to the rest of the world, these insights are useful to better understand fundamental biological processes of gamma-herpesvirus infection and their epidemiological characteristics to inform appropriate therapeutic interventions and to control their transmission. Lastly, given there are no equivalent datasets from African populations, summary statistics derived from this study are a useful resource for others to use.

## Methods

**Ethics approval and consent**. Informed consent was obtained from all participants either in conjunction with parental/guardian consent for under 18-year-olds with signature, or a thumb print if the individual was unable to write. The study was approved by the Uganda Virus Research Institute, Research Ethics committee (UVRI-REC) (Ref. GC/127/10/10/25), the Uganda National Council for Science and Technology (UNCST) (Ref. HS 870), and the UK National Research Ethics Service, Research Ethics Committee (UK NRES REC) (Ref. 11/H0305/5).

**Sample selection and collection**. The GPC is a population-based cohort in Kyamulibwa, rural south-west Uganda, consisting of 25 neighbouring villages inhabited mainly by farmers who grow subsistence crop, cultivate coffee for trade and also raise livestock[26,64]. Households are scattered with some concentrated in the trading centres. Villages were categorised by urbanicity quartiles to reflect shared socio-demographic and 'urban' characteristics based on seven components: population size, economic activity, built environment, communication services, educational facilities, health services, and diversity as previously described elsewhere[27]. Blood samples from 7000 GPC study participants, representing 11 self-reported ethnolinguistic groups, were collected during medical survey sampling conducted in the study area between 2010 and 2011[26]. Details of sexual behaviour, medical, socio-demographic and geographic factors were also recorded. Serum was tested for HIV-1, Hepatis B (HBV) and Hepatitis C (HCV) infections and the remainder was stored at −80 °C in freezers in Entebbe prior to further serological testing.

**Serology and quality control of phenotypic data**. We quantified MFI to measure the levels of serum IgG antibodies to the KSHV encoded antigens ORF73, K8.1 and K10.5; and to the EBV encoded EBNA-1 VCA and EAD covalently bound to fluorescent magnetic beads and read on a multiplexed platform[28]. Pairwise-correlation between phenotypes was tested for using Pearson's correlation in R. For each virus, seropositivity was defined as having antibody level to any antigen tested above a prespecified cut-off, as previously described[29]. Of the original ~7000 people genotyped, we were able to link phenotype results from 4365 people (mean age ± SD = 34 ± 18 years, 54% female) from samples collected from the GPC in 2011.

**Statistical analysis of quantitative antibody levels**. To investigate factors influencing IgG antibody response to KSHV and EBV infections, categorical outcome variables for antibody serostatus (i.e. positive/negative)[29] and continuous variables for antibody levels (log₁₀ transformed MFI) were generated and analysed with potential risk factors. The following variables were considered to be possible risk factors or confounders: sex, age (years) or age group (categorised as 13–18, 19−35 or ≥40 years), ethnic group (Baganda or other), education level attained (none/primary, junior primary, senior secondary or tertiary), village urbanicity quartile (1−4 with 4 being most urban) based on shared socio-demographic or 'urban' characteristics as previously described[27]. HIV infection, HBV infection, HCV infection, and anti-KSHV serostatus and log₁₀ antibody levels (anti-K8.1, anti- ORF73 and anti-K10.5), or anti-EBV antibody serostatus and antibody levels (anti-EBNA-1, anti-VCA and anti-EAD log₁₀ MFI). Multivariate logistic or linear regression models were fitted to examine variables predictive of antibody levels. All variables in the model were independent of each other except for urbanicity quartile and education ($\chi^2$, $p < 0.001$) and age and education ($\chi^2$, $p < 0.001$); nonetheless, we do not identify significant collinearity in our models as variance inflation factors were all close to 1. A multiple testing $p$ value of <0.006 was used to determine statistical significance.

**Genetic data curation and quality control**. Of the 7000 GPC samples, 5000 were densely genotyped on the Illumina HumanOmni 2.5M BeadChip array and we then imputed additional variants into the genotype chip dataset using a merged 1000 Genomes phase 3 [65], African genome variation project [32] and UG2G (Uganda 2000 Genomes) reference panel in IMPUTE2 [66]. Whole-genome sequencing was performed on 2000 samples with 100 base paired end sequencing at 4× coverage on the Illumina HiSeq 2000 platform following the manufacturer's protocol. Stringent variant and sample quality control (QC) filtering was performed. Low-quality variants that mapped to multiple regions within the human genome or did not map to any region, and duplicate variants genotyped on the chip were removed. We excluded samples with a call rate <97% and heterozygosity >3 SD from the mean, discordant genetic sex and reported sex, and sites deviating from Hardy−Weinberg equilibrium ($p < 10^{-8}$). Following imputation, we only included high-quality sites (info score > 0.3 and $r^2 > 0.6$) with minor allele frequency (MAF) ≥ 0.5%. We also removed samples without matching phenotype and genotype or sequence data. Of the merged datasets, 343 samples had overlapping genotype and sequence variant calls with a final concordance of 98% was achieved for all SNPs. The merged datasets post QC filtering resulted in 4365 samples with KSHV and EBV phenotypes and ~17 M SNPs across the autosomes and X-chromosome for analyses. For EBV analyses we first performed primary GWAS of 3289 unique individuals and then in 4365 individuals i.e. in combination with 1076 overlapping samples present in our pilot study [11].

**HLA region imputation**. Human leucocyte antigen alleles were imputed using genotype data available from across the extended MHC region only (chromosome 6 base pair positions 25,500,000−34,000,000 in genome build 37) and SNP strandedness was checked with the HLA*IMP front-end software [67]. HLA genotypes of 117 alleles (AF > 1%), at 11 loci were imputed with HLA*IMP:02 [63] using the standard HLA*IMP:02 multi-ethnic reference graphs, representing ~800 reference individuals for HLA-DRB3/4/5 and >6000 reference individuals for all other loci.

**Heritability of antibody response traits in the GPC**. Narrow-sense heritability ($h^2$) for anti-EBNA-1 IgG, anti-VCA IgG, anti-ORF73 IgG, anti-K8.1 IgG and anti-K10.5 IgG traits were estimated using a linear mixed model (LMM) in FaST-LMM with two random effects, one based on genetic effects and the other on environmental effects using spatial location [31] recorded as Global Position System (GPS) coordinates as a proxy for environmental effects.

**Genome-wide association analyses**. The statistical power to identify genetic variants of genome-wide significance and with different effect sizes given the sample size was estimated using QUANTO software (http://biostats.usc.edu/software). We conducted analyses for all quantitative antibody traits for KSHV ($N = 4365$) and EBV ($N = 3289$ and $N = 4365$ (including overlapping samples from pilot study) by first applying a multivariable linear regression model adjusting for significant covariates in R. Residuals of MFI values used for analyses were then transformed using inverse, rank-based normalisation in R to ensure a standard normal distribution for the phenotypes and confirmed by visualisation. To control for cryptic relatedness and population structure within the GPC, the GWAS of transformed phenotypes was performed using the standard mixed model approach in GEMMA [68]. A kinship matrix to define pairwise genetic relatedness among individuals was generated using pooled imputed genotypes and sequence data for all autosomes and X-chromosome using the $k = 1$ option in GEMMA. The data were LD-pruned ($r^2 = 0.2$) using dosages and an MAF threshold of 1% was applied. Environmental correlation using GPS data as a proxy, genotyping or sequencing method were also adjusted for as additional covariates during analysis in GEMMA. To identify distinct SNPs, conditional analysis was performed in GEMMA. Each SNP within 1 MB of the lead association SNP was conditioned. If any SNP was statistically significant, it was added stepwise onto the mixed model and analysed jointly; this was done until no SNPs with $p < 5 \times 10^{-9}$ [11,33] remained. All SNPs remaining statistically significant were considered distinct association signals. To functionally annotate our most significant associations, we used the Ensembl Variant Effect Predictor (VEP) [69] and the gene/tissue expression database (GTEx; https://gtexportal.org) V8 release to access data on expression quantitative trait loci (eQTLs) from tissues. To test associations between imputed HLA allele and amino acid dosages and phenotypes, we performed linear regression of transformed phenotypes in GEMMA as performed in the SNP-based GWAS for that phenotype.

**Trans-ancestry meta-analysis and fine mapping**. MANTRA was used to perform a genome-wide trans-ethnic meta-analysis for anti-EBNA IgG responses with association summary statistics of 4365 individuals from our Ugandan GWAS combined with publicly available data of 1000 Genomes-imputed European ancestry GWAS from 2162 seropositive individuals, giving a total of 6527 individuals with ~4.6 million shared SNPs for analysis. The MANTRA approach leverages differences in LD structures across populations to account for differences in genetic architecture and accommodates heterogeneity of allelic effects between distantly related populations within a Bayesian partition framework [37]. To determine statistical significance, we use a threshold of $\log_{10}$ Bayes Factor (BF) > 6 which is comparable to a $p < 5 \times 10^{-8}$, previously determined by Wang et al. [38]. Heterogeneity of allelic effect sizes was calculated using Cochran's $Q$-test for heterogeneity in METAL [70]. Using MANTRA results we generated 99% credible sets most likely

to drive association signals and contain causal variants (or tagging unobserved causal variants) and compared fine-mapping intervals for each associated lead SNP by analysing the variants 500 kb up- and downstream of the lead SNP in the Ugandan and combined Ugandan + European datasets. For this, posterior probabilities were calculated for SNPs and then ranked in decreasing order according to BF, proceeding down the rank until the cumulative posterior probability exceeded 99% [39,40]. All SNPs ≥ 0.99 were included in the credible sets.

**Reporting summary**. Further information on research design is available in the Nature Research Reporting Summary linked to this article.

## Data availability

Summary statistics are available on the NHGRI-EBI GWAS Catalogue https://www.ebi.ac.uk/gwas/downloads/summary-statistics under accession codes: GCST90000522, GCST90000523, GCST90000524, GCST90000525, GCST90000526.

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

## Acknowledgements

We thank all study participants who contributed to this study. We acknowledge the African Genome Variation Project (AGVP) for sharing data resources to contextualise our results. The GPC is jointly funded by the UK Medical Research Council (MRC) and the UK Department for International Development (DFID) under the MRC/DFID Concordat agreement. Further funding was obtained from the Wellcome Trust (WT098051 and WT090132), the UK Medical Research Council and with federal funds from the National Cancer Institute, National Institutes of Health, under Contract No. HHSN261200800001E and Contract 75N91019D00024. D.G. received a UKRI/Rutherford HDR-UK fellowship, serial number: MR/S003711/1.

## Author contributions

N.S., I.B., P.K., M.S., D.W. and R.N. designed the study, interpreted the results and wrote the manuscript. W.M., N.L. and D.W. performed phenotype assay development and validation. T.C., S.F., M.O.P. and D.G. carried out curation of sequence and genotype data, including establishment of the reference panel including the African Genome Variation Project data.

N.S. carried out all statistical analyses: regression, heritability and genetic association analyses. A.T.D. and A.J.M. performed HLA imputation of genotypes and assisted with interpretation of findings. V.M., E.M.C.C. and M.L.H. contributed to revising of the manuscript. G.A. was the programme leader for the GPC. C.P. and E.H.Y. were the scientific coordinators of the project and coordinated regulatory approvals. All authors commented on the interpretation of the results, reviewed and approved the final manuscript.

## Competing interests

The authors declare no competing interests.
