## [Peer Review File · Nature Communications]

Editorial note: in the second round of review, reviewer 2 checked the authors' response to reviewer 3's comments.

Reviewers' comments:

Reviewer #1 (Remarks to the Author):

Sallah and a team of very experienced epidemiologists and genomicists present results from a study of Ugandan adults participating in a population cohort study in which they assess the association between sociodemographic and genetic factors and serological response to either EBV (HHV4) and KSHV (HHV8). They combined survey data that obtained sociodemographic data, with whole-genome sequencing or genotyped and imputed data and serological testing of individuals to the above viruses as well as HCV, HBV and HIV. They perform regression analyses for several sociodemographic factors, a heritability analysis and a genome-wide association study. Their work's strengths are the substantial sample size, unique population and robust genotyping efforts.

Their work would be greatly strengthened by some changes however:

1. They perform an analysis of several different serological responses to KSHV antigens(ORF73, K10.5 and K8.1) and define overall 'KSHV positivity' as a positive result to any one antigen, concordant with other studies. They go on to present analyses against individual antigens as well as for KSHV seropositive individuals en masse. It is arguable whether there is much biological sense in analysing each antibody response independently: this requires some justification; to my mind the primary analysis should be KSHV seropositivity esp. for the GWAS analyses; and in sensitivity analyses one could explore the various antibodies. If the authors can justify presenting individual antigen analyses then, in order to interpret how independent these analyses are (if at all), it is necessary to present the correlation between the measures (for example even in a Euler/venn diagram).
2. The authors perform analyses of antibody titre, but it seems (as the methods do not make it clear) that these analyses include those who are seronegative hence largely recapitulating what is presented when comparing seronegative and seropositive individuals.
3. The cohort and exposure variables are described in very cursory form. For example, it is mentioned that it is a rural cohort, yet the analysis presents quartile of urbanicity. It is hard for a reader to calibrate the analysis of urbanicity without appropriate definitions of this: presumably quartile 4 urbanicity in the GPC does not approximate a setting like Entebbe?
4. It is not clear whether the analyses presented for the environmental determinants are from multivariate or univariate analyses. The current supplementary table doesn't show the univariate and multivariate estimates, and it is not clear what (if anything) is being adjusted for here. The reason this is important is because there is likely to be significant collinearity for example the individuals other than Baganda which include refugees from Rwanda and elsewhere are much more likely to have lower urbanicity, lower education etc. The text is somewhat hard to follow and replacing it by a table may be advisable for example lines 125-127 seem to repeat the result reported in line 119.
5. It is not clear whether any batch correction was applied for the antibody titre analyses
6. For the heritability analyses if the authors elect to present individual antigens, then a coheritability analysis may be helpful
7. Figure 1 Shows the proportion of participants in the cohort seropositive for 1,2,3 or 4 infections. I am not sure how valuable this is and would encourage this be replaced or complemented by a figure showing the two specific antigens for the viruses that this paper focuses on viz. KSHV and EBV
8. Figure 2+3 could be enhanced by reporting it a like a forest plot: with OR and 95% CI reported for ease of reference.
9. In the primary genetic GWAS analysis the findings are significantly weakened by the absence of any replication effort, any functional analyses (although the methods mention in silico eQTL analyses these are not presented) or any fine mapping effort. The authors claim that HLA imputation was not performed because of an absence of reference panels is not well founded: there are now several African reference panels available, including ones the authors are developing and have presented. Because many of the main findings are in the HLA locus, this is really

important.

10. The claim that the KSHV association as 'far weaker' (line 358) than the EBV association is not well founded, indeed the OR's are very similar for the peak KSHV association and the EBV association but the p values vary in accordance with the differential power due to differences in allele frequency.

11. The title of the paper doesn't seem to match up to the data well.

12. It would be helpful for the authors to discuss their sociodemographic analyses: for example it is surprising that increasing age is associated with lower EBV seroprevalence; in most other settings in the world the opposite is seen.

Reviewer #2 (Remarks to the Author):

Summary: The authors describe a study investigating the environmental and genetic determinants of serological response to EBV and KSHV, two high prevalence gamma herpesviruses associated with malignancies, in a general population sample of >4,000 individuals in Africa. The authors report on several environmental factors associating with seropositivity and antibody levels including confection, sex and other demographic factors. The authors observe significant heritability associated with antibody response to both pathogens which is mitigated upon correction for environmental factors. Genetic variant level analysis confirms the strong impact of HLA class II alleles on EBV serology with only suggestive associations for KSHV.

Overall, this is an informative and well executed study of environmental and genetic factors contributing to the immune response to globally significant pathogens. However, I feel additional detail should be provided to improve clarity and interpretability of the findings as outlined below.

1. For the environmental analysis, it is unclear which of the associated factors are independently associated with serostatus. Are the reported results from multiple regression or single variables? The authors mention that the associated variables have adjusted p-values below $P < 0.05$ but it is unclear what adjustment was used. This should be made explicit.

2. To my mind, the authors are trying to convey too much information in figures 2 and 3. I recommend splitting by pathogen (KSHV or EBV) and making these into 2-panel figures. I also recommend including point estimates for all antibodies tested for each variable so that each section contains the same information in the same order and one doesn't have to continually reference the colour scheme.

3. Also for figure 2, it is not well explained how the KSHV and EBV seropositivity phenotype is defined. Is it by positivity to any antigen or some sort of cumulative score?

4. The drop in sample number for the EBV genetic analysis should be better described. My understanding is that this excludes 1,076 individuals from their previous publication. If this is the case it should be stated in the relevant results section to avoid confusion. Also, I would think a combined analysis of the present sample and the pilot study would be a useful contribution to provide the analysis with the maximum power.

5. In the genetic analysis, the authors use GPS coordinates to correct for environmental variables. Although this approach is useful, my understanding is that it is most applicable when detailed data are not available. However, in the previous sections the authors identified several environmental factors that were directly assessed and shown to influence the phenotypes. I feel it would be more appropriate to control for those significant environmental factors directly rather than relying on GPS coordinates. At a minimum it would be important to demonstrate that correction for significant environmental factors doesn't materially change the findings.

6. Page 14 line 209 states "In this cohort, 72% and 94% of individuals were categorized as KSHV and EBV seropositive, respectively." These numbers are out of line with what is reported in Table 1 (91.3% KSHV; 90.6% EBV) and requires explanation.

7. For the KSHV GWAS, although I do think it is appropriate to include results meeting the traditional GWAS threshold of $P < 5 \times 10^{-8}$, particularly given the plausibility of HLA associations, I am generally less enthusiastic about the 'suggestive significance' level of $P < 1 \times 10^{-6}$ as this no doubt includes several false positive associations. I recommend removing that comment in lieu of a statement of availability of the full association results.

8. The number of previously reported KS variants the authors were able to assess seems low (8/31) given the density of genotyping and imputation in this sample. Can the authors provide an explanation as to why these variants were not detected and was any attempt made to test proxies for the missing variants?

9. For completeness, Manhattan plots for the anti-EBV antibody GWAS should be included either in the main text or the supplement. Also, there were presumably no non-HLA associations detected in the analysis? If so I think this should be stated.

Minor points:

In the introduction and discussion, the authors write: "We and others have reported associations in the HLA class II region with EBV EBNA-1 antibody response" but only provide a single reference. Reference to the work by the "and others" would be appropriate here.

Given the results first organization of the manuscript, some methodological details should be included in each section. For example GPC should be defined with a brief description provided in the first results section to improve readability. Also, brief descriptions of the statistical methods used (single vs multiple regression, covariates included etc) would be helpful.

Paul J McLaren, PhD

Reviewer #3 (Remarks to the Author):

Sallah et al. studied the factors associated with humoral immune response to infection by γ -herpesviruses in a large African cohort. The EBV analyses confirmed and expanded the known associations in the HLA class II region, while the KSHV analyses did not reveal any significant association after correction for multiple testing. A large African-based GWAS of the immune response to prevalent viral diseases is welcome. However, the paper suffers from a number of weaknesses.

Major comments:

1. Correlation and causation are often confounded in the way the results are presented. This study mostly shows associations and the wording should reflect this reality. For example, in the first paragraph of the discussion (line 282), "..., where we found HIV infection in particular influencing IgG antibody response levels to KSHV and EBV infection. It is interesting that while HIV infection lowered antibody responses to KSHV, the inverse was observed for EBV antibodies.": HIV infection is highly unlikely to influence antibody responses against EBV or KSHV in such a manner, but is rather an indirect epidemiological marker.

2. Correction for multiple testing should be stringent and only significant results should be discussed at length. In the Results, the authors rightly state: "Correcting for multiple testing and accounting for the lower linkage disequilibrium (LD) in African populations the genome-wide

significance threshold was adjusted to $p < 5 \times 10^{-9}$ ". However, they immediately follow by describing multiple non-significant associations. These "not-quite-significant" associations are presented again in the discussion. This is not only confusing to the reader, it also gives the impression that the authors don't really know what should be done about these results. Is there any reason to believe that the reported polymorphisms are worth investigating further, in which case they should indeed be mentioned in the main text of the paper? If not, they should only be available as part of the summary statistics of the study, without further emphasis.

3. Results line 209: "In this cohort, 72% and 94% of individuals were categorized as KSHV and EBV seropositive". Why only 72% of KSHV seropositivity now? It is >90% in the first paragraph of the results.

4. Methods: it is not clear which method was used for which genome-wide analysis. There are two contradictory statements in the "genome-wide analyses" section: "We conducted analyses for all quantitative antibody traits for KSHV (N=4,365) and EBV (N=3,289) applying a linear regression model adjusting for age, age² and sex in R" and "To control for cryptic relatedness and population structure within the GPC, the GWAS was performed using the standard mixed model approach in GEMMA".

5. Multiple linguistic imprecisions and grammatical errors make some sections of the manuscript hard to follow. This should be corrected. Some examples:

- Intro lines 44-45: "KSHV displays striking geographic variation that parallels KS incidence of disease caused by the virus". Please rephrase

- Intro lines 53 to 55: "For both viruses, B-lymphocytes act as a reservoir of latent infection and have been found to promote latency by subverting the host immune response and inhibiting lytic reactivation of each other in dually infected cells".

- Discussion line 311: "Here, we used IgG response traits as an intermediate phenotype for KS, seeing as sample sizes are limiting to conduct well powered GWASs for KSHV-associated diseases such as KS, and that previous studies have shown correlation of IgG levels with the development of KS."

Minor comments:

1. Abstract: "study" is missing in this sentence: "In 4,365 individuals from an African population cohort, we performed epidemiological analysis and genome-wide association (GWAS)..."

2. Results: the global serology prevalence results are presented repeatedly, i.e. in Table 1, Figure 1 and in the text. This should be streamlined.

3. Still in the Results: the long paragraphs "Environmental determinants of IgG serostatus to KSHV and EBV infections" and "Environmental determinants of IgG response levels to KSHV and EBV infections" are hard to read and are mostly descriptions of Figure 2 and Figure 3, respectively. They should be shortened to only highlight essential results. Figure 2 should also be redrawn to make it more informative (most ORs are very close to 1, so that part of the figure should be larger).

4. Methods, line 426: "Of the merged datasets, 343 samples had overlapping genotype and sequence variant calls for which a final concordance of 93.1% was achieved for all SNPs." This concordance rate sounds quite low. Please comment.

Reviewers' comments:

Reviewer #1 (Remarks to the Author):

Sallah and a team of very experienced epidemiologists and genomicists present results from a study of Ugandan adults participating in a population cohort study in which they assess the association between sociodemographic and genetic factors and serological response to either EBV (HHV4) and KSHV (HHV8). They combined survey data that obtained sociodemographic data, with whole-genome sequencing or genotyped and imputed data and serological testing of individuals to the above viruses as well as HCV, HBV and HIV. They perform regression analyses for several sociodemographic factors, a heritability analysis and a genome-wide association study. Their work's strengths are the substantial sample size, unique population and robust genotyping efforts.

Their work would be greatly strengthened by some changes however:

1. They perform an analysis of several different serological responses to KSHV antigens (ORF73, K10.5 and K8.1) and define overall 'KSHV positivity' as a positive result to any one antigen, concordant with other studies. They go on to present analyses against individual antigens as well as for KSHV seropositive individuals en masse. It is arguable whether there is much biological sense in analysing each antibody response independently: this requires some justification; to my mind the primary analysis should be KSHV seropositivity esp. for the GWAS analyses; and in sensitivity analyses one could explore the various antibodies. If the authors can justify presenting individual antigen analyses then, in order to interpret how independent these analyses are (if at all), it is necessary to present the correlation between the measures (for example even in a Euler/venn diagram).

Thank you for the comment and apologies for the lack of clarity. A susceptibility GWAS (seropositive vs seronegative) would not be appropriate as most individuals in the cohort are seropositive (>90%) and presumably infected from childhood (Newton et al., 2018). A quantitative analysis provides a greater dynamic range that highlights inter-individual variability and thus allows for greater power considering sero-reactivity to different antigens can develop at different time in the natural history of infection, potentially representing, for example, different stages of the viral life cycle i.e. latent vs lytic (117-119). While the responses are modestly phenotypically correlated (r^2 ranging from 0.12-0.43), they are likely not genetically correlated (i.e. influenced by the same genetic loci) and thus could have different implications with regards to infection and response. We have added this statement in the results (lines 124-125) and we have added in a correlation matrix of phenotypes in Supplementary File 2 Figure S2 for clarity.

Reference:

Newton R, Labo N, Wakeham K, Miley W, Asiki G, Johnston T, Whitby D. Kaposi's sarcoma associated herpesvirus in a rural Ugandan Cohort: 1992-2008. *J Infect Dis*, 2018; 217(2): 263-269.

2. The authors perform analyses of antibody titre, but it seems (as the methods do not make it clear) that these analyses include those who are seronegative hence largely recapitulating what is presented when comparing seronegative and seropositive individuals.

As mentioned above, most individuals in the cohort are seropositive (>90%) and presumably infected from childhood (Newton et al., 2018) those who present as seronegative are also likely to have been exposed and thus antibodies produced are very low/undetectable, as such they are also informative and useful to include in analysis (lines 196-199). In this case, a quantitative analysis utilising MFI (not actual titres) provides a greater dynamic range that highlights inter-individual variability in strength of antibody response and thus allows for greater power. Excluding individuals who are seronegative (<10%) doesn't change our findings.

3. The cohort and exposure variables are described in very cursory form. For example, it is mentioned that it is a rural cohort, yet the analysis presents quartile of urbanicity. It is hard for a reader to calibrate the analysis of urbanicity without appropriate definitions of this: presumably quartile 4 urbanicity in the GPC does not approximate a setting like Entebbe?

Thank you for raising this important point. This cohort is indeed rural and does not approximate a setting like Entebbe, however urbanicity quartiles were used to group villages by shared socio-demographic or 'urban' characteristics and given scores on a scale of 1-32 to reflect this as previously described for the GPC in Riha et al, 2014: <https://doi.org/10.1371/journal.pmed.1001683> . A marked variation in levels of urbanicity across the villages, largely attributable to differences in economic activity, civil infrastructure, and availability of educational and healthcare services has been previously described. We have now added the following statement in the methods to improve clarity "Villages were grouped into urbanicity quartiles to reflect shared socio-demographic and 'urban' characteristics and based on seven components: population size, economic activity, built environment, communication services, educational facilities, health services, and diversity. Quartiles ranged from 1 (no educational facilities and no households with electricity, and the average number of years that women spent in education was less than 6 y) to 4 (14.8% of households had electricity; at least one nursery, primary, or secondary school in the village; and women on average spent at least 6 y in education) as previously described elsewhere" (lines 96 -100 and 436-439)

4. It is not clear whether the analyses presented for the environmental determinants are from multivariate or univariate analyses. The current supplementary table doesn't show the univariate and multivariate estimates, and it is not clear what (if anything) is being adjusted for here. The reason this is important is because there is likely to be significant collinearity for example the individuals other than Baganda which include refugees from Rwanda and elsewhere are much more likely to have lower urbanicity, lower education etc. The text is somewhat hard to follow and replacing it by a table may be advisable for example lines 125-127 seem to repeat the result reported in line 119.

We have made adjustments in the text to improve clarity. As now indicated in the methods (lines 468-472), the analyses are multivariate estimates adjusted for age and sex where appropriate, see statement "Multivariate logistic or linear regression models were fitted to examine variables

predictive of antibody levels. All variables in the model were independent of each other except for urbanicity quartile and education (χ^2 $p < 0.001$) and age and education (χ^2 $p < 0.001$), nonetheless, we do not identify significant collinearity in our models as variance inflation factors were all close to 1. A multiple testing p -value of < 0.006 was used to determine statistical significance.”

5. It is not clear whether any batch correction was applied for the antibody titre analyses.

No, we didn't perform batch corrections. Reproducibility is very high for the selected analytes, including repeatability and no batch effect was observed (the coefficient of variation (CV) in the positive control antigens ($n = 50$ experiments) ranged from 1.8% to 4.0% in \log_{10} transformed MFI values). All assays were performed by two operators using robotic handling as described previously and cited in the main text (Labo et al 2014, and Newton et al 2018).

References included here for ease:

Labo N, et al. Heterogeneity and breadth of host antibody response to KSHV infection demonstrated by systematic analysis of the KSHV proteome. *PLoS Pathog.* 2014;10(3):e1004046. Epub 2014/03/29. doi: 10.1371/journal.ppat.1004046. PubMed PMID: 24675986

Newton R, et al. Kaposi Sarcoma-Associated Herpesvirus in a Rural Ugandan Cohort, 1992-2008. *J Infect Dis.* 2018;217(2):263-9. Epub 2017/11/04. doi: 10.1093/infdis/jix569. PubMed PMID: 29099933

6. For the heritability analyses if the authors elect to present individual antigens, then a coheritability analysis may be helpful

Thank you for the suggestion. We have not performed a coheritability (i.e. genetic correlation) as we don't meet the sample size criteria of > 5000 to be able to report reliable findings as described in Bulk-Sullivan et al, 2015: <https://doi.org/10.1038/ng.3211>). There is a result of the caveat that this will not reliably account for regression LD, or relatedness between individuals and thus will be unable distinguish between polygenicity and population stratification.

7. Figure 1 Shows the proportion of participants in the cohort seropositive for 1,2,3 or 4 infections. I am not sure how valuable this is and would encourage this be replaced or complemented by a figure showing the two specific antigens for the viruses that this paper focuses on viz. KSHV and EBV

Thank you for the suggestion. Figure 1 has now been replaced with box plots showing antibody distributions for both infections.

8. Figure 2+3 could be enhanced by reporting it a like a forest plot: with OR and 95% CI reported for ease of reference.

We have now converted beta effect sizes to ORs and split Figure 2 and 3 by pathogen (i.e. Fig 2 KSHV, Fig 3 EBV).

9. In the primary genetic GWAS analysis the findings are significantly weakened by the absence of any replication effort, any functional analyses (although the methods mention in silico eQTL analyses these are not presented) or any fine mapping effort. The authors claim that HLA imputation was not performed because of an absence of reference panels is not well founded: there are now several

African reference panels available, including ones the authors are developing and have presented. Because many of the main findings are in the HLA locus, this is really important.

This EBV study is an extension of pilot results from our EBV GWAS (Sallah et al, 2017) and thus a replication of previous findings (lines 254-256), for KSHV we are not aware of any other datasets with relevant data so are not able to perform replication analyses. In silico eQTL analyses has now been included in the results (lines 230-235). Although we are not able to use the HLA imputation reference panel from African data being developed by some of the authors as it is under embargo, we have now performed HLA imputation using the standard HLA*IMP:02 multi-ethnic reference graphs, representing ~800 reference individuals for HLA-DRB3/4/5 and >6000 reference individuals for all other loci (lines 266-272 and 494-500) . We have also now included results of fine-mapping by trans-ethnic meta-analysis of our Ugandan dataset plus published European ancestry GWAS of EBV EBNA-1 antibody responses and generation of credible sets using MANTRA (lines 285-302 and 535-552).

10. The claim that the KSHV association as 'far weaker' (line 358) than the EBV association is not well founded, indeed the OR's are very similar for the peak KSHV association and the EBV association but the p values vary in accordance with the differential power due to differences in allele frequency.

We agree with the reviewer and have now removed this statement from the text.

11. The title of the paper doesn't seem to match up to the data well.

We respectfully disagree with the reviewer on this point, and as none of the other reviewers or the editor have suggested a change, we have elected to retain the original submitted title. However, if the reviewers and editor feel strongly, we could consider alternative titles.

12. It would be helpful for the authors to discuss their sociodemographic analyses: for example, it is surprising that increasing age is associated with lower EBV seroprevalence; in most other settings in the world the opposite is seen.

Thank you for the comment we have now discussed this finding in lines 315-319 and here: While we only observed modest effects (OR=0.64) in the categorical age group 40+ in comparison to 13-18y, it is worth noting that this cohort is largely consisting of adults (>13years, mean \pm SD = 34 \pm 18 years) and thus, is less comparable with previously published studies that largely include children. In addition, the background antigenic milieu in this population is totally different to western populations, it is likely that while antibody levels increase in childhood, they could decay with time (in adulthood), whilst the immune system is busy coping with other things as observed in other studies for other viral infections (e.g. CMV - Stockdale L, Nash S, Nalwoga A, Painter H, Asiki G, Fletcher H, Newton R. Cytomegalovirus epidemiology and relationship to cardiovascular and tuberculosis disease risk factors in a rural Ugandan cohort. PLoS ONE, 2018; 13(2): e0192086.) or as a result of increased exposure to other infections. Age as a continuous variable was not associated with differences in EBV antibody levels (Supplementary Table S3).

Reviewer #2 (Remarks to the Author):

Summary: The authors describe a study investigating the environmental and genetic determinants of serological response to EBV and KSHV, two high prevalence gamma herpesviruses associated with malignancies, in a general population sample of >4,000 individuals in Africa. The authors report on several environmental factors associating with seropositivity and antibody levels including confection, sex and other demographic factors. The authors observe significant heritability associated with antibody response to both pathogens which is mitigated upon correction for environmental factors. Genetic variant level analysis confirms the strong impact of HLA class II alleles on EBV serology with only suggestive associations for KSHV.

Overall, this is an informative and well executed study of environmental and genetic factors contributing to the immune response to globally significant pathogens. However, I feel additional detail should be provided to improve clarity and interpretability of the findings as outlined below.

1. For the environmental analysis, it is unclear which of the associated factors are independently associated with serostatus. Are the reported results from multiple regression or single variables? The authors mention that the associated variables have adjusted p-values below $P < 0.05$ but it is unclear what adjustment was used. This should be made explicit.

Thank you for the comment and apologies for the lack of clarity. The analyses are multivariate estimates adjusted for age and sex where appropriate. All variables in the model are independent of each other except for urbanicity and education (χ^2 $p < 0.001$), nonetheless, we do not identify significant collinearity in our models as variance inflation factors are all close to 1 and thus, we have chosen to keep them both in the model. This has now been added to the text for clarity see statement (and lines 469-472): "Multivariate logistic or linear regression models were fitted to examine variables predictive of antibody levels. All variables in the model were independent of each other except for urbanicity quartile and education (χ^2 $p < 0.001$) and age and education (χ^2 $p < 0.001$), nonetheless, we do not identify significant collinearity in our models as variance inflation factors were all close to 1. A multiple testing p-value of < 0.006 was used to determine statistical significance"

2. To my mind, the authors are trying to convey too much information in figures 2 and 3. I recommend splitting by pathogen (KSHV or EBV) and making these into 2-panel figures. I also recommend including point estimates for all antibodies tested for each variable so that each section contains the same information in the same order and one doesn't have to continually reference the colour scheme.

Figures 2 and 3 have now been adjusted with suggestions incorporated to improve clarity.

3. Also for figure 2, it is not well explained how the KSHV and EBV seropositivity phenotype is defined. Is it by positivity to any antigen or some sort of cumulative score?

Apologies for the lack of clarity. Serostatus definition is described in the results and methods, it is seropositivity (i.e seroreactive levels above specific cutoffs) to any of the antibodies tested above pre-established cutoffs as previously derived and published elsewhere (see references below). This is now reiterated as appropriate in the figures 2 and 3, methods and results section for clarity, see statements below:

In results (lines 103-105): “In this study, 91% of individuals were categorized as seropositive for EBV based on detectable IgG levels against either -EBNA-1 or VCA and 91% categorised as seropositive for KSHV based on detectable IgG levels against either ORF73, K10.5 or K8.1 over pre-established cut-offs”

In methods (lines 451-453): “For each virus, seropositivity was defined as having antibody level to any antigen tested above a prespecified cut-off, as previously described”

References:

1. Labo N, et al. Heterogeneity and breadth of host antibody response to KSHV infection demonstrated by systematic analysis of the KSHV proteome. *PLoS Pathog.* 2014;10(3):e1004046. Epub 2014/03/29. doi: 10.1371/journal.ppat.1004046. PubMed PMID: 24675986

2. Newton R, et al. Kaposi Sarcoma-Associated Herpesvirus in a Rural Ugandan Cohort, 1992-2008. *J Infect Dis.* 2018;217(2):263-9. Epub 2017/11/04. doi: 10.1093/infdis/jix569. PubMed PMID: 29099933

4. The drop in sample number for the EBV genetic analysis should be better described. My understanding is that this excludes 1,076 individuals from their previous publication. If this is the case it should be stated in the relevant results section to avoid confusion. Also, I would think a combined analysis of the present sample and the pilot study would be a useful contribution to provide the analysis with the maximum power.

Thanks for this suggestion. We have now combined the analysis of our pilot and current study and present both the primary analysis (N=3289, which is an independent replication of our previous results) and the results in the total 4,365 individuals. See lines 256-260 and 512-519.

5. In the genetic analysis, the authors use GPS coordinates to correct for environmental variables. Although this approach is useful, my understanding is that it is most applicable when detailed data are not available. However, in the previous sections the authors identified several environmental factors that were directly assessed and shown to influence the phenotypes. I feel it would be more appropriate to control for those significant environmental factors directly rather than relying on GPS coordinates. At a minimum it would be important to demonstrate that correction for significant environmental factors doesn't materially change the findings.

Thanks for raising this important point. Previous studies have reported other environmental factors which we have not measured, such as parasitic infections influencing antibody response as mentioned in the discussion: “As Uganda is also a malaria endemic area and studies have reported co-infection with *P. falciparum* and other parasites as having an influence on antibody titre...” (see some References below). Thus, we use GPS coordinates to capture missing covariates and potential unmeasured confounding related to location for the heritability analysis. In the GWAS, we only corrected for significant environmental covariates to avoid over correction.

References:

1. Wakeham K, et al. Parasite infection is associated with Kaposi's sarcoma associated herpesvirus (KSHV) in Ugandan women. *Infect Agent Cancer.* 2011;6(1):15. doi: 10.1186/1750-9378-6-15. PubMed PMID: 21962023; PubMed Central PMCID: PMC3197512.

2. Nalwoga A, et al. Association between malaria exposure and Kaposi's sarcoma-associated herpes virus seropositivity in Uganda. *Trop Med Int Health.* 2015;20(5):665-72. doi: 10.1111/tmi.12464. PubMed PMID: 25611008; PubMed Central PMCID: PMC390463.

3. Newton R, et al. Determinants of Gammaherpesvirus Shedding in Saliva Among Ugandan Children and Their Mothers. *J Infect Dis.* 2018;218(6):892-900. Epub 2018/05/16. doi: 10.1093/infdis/jiy262. PubMed PMID: 29762709; PubMed Central PMCID: PMC6093317

6. Page 14 line 209 states “In this cohort, 72% and 94% of individuals were categorized as KSHV and EBV seropositive, respectively.” These number are out of line with what is reported in Table 1 (91.3% KSHV; 90.6% EBV) and requires explanation.

Thank you for pointing out this discrepancy. This is an error and has been corrected in the text, see statement in results (lines 103-105): “In this study, 91% of individuals were categorized as seropositive for EBV based on detectable IgG levels against either -EBNA-1 or VCA and 91% categorised as seropositive for KSHV based on detectable IgG levels against either ORF73, K10.5 or K8.1 over pre-established cut-offs” and in discussion (lines 312-313) : “In the GPC, prevalence estimates of KSHV and EBV are high (>90%) and consistent with previous findings in Uganda”

7. For the KSHV GWAS, although I do think it is appropriate to include results meeting the traditional GWAS threshold of $P < 5 \times 10^{-8}$, particularly given the plausibility of HLA associations, I am generally less enthusiastic about the ‘suggestive significance’ level of $P < 1 \times 10^{-6}$ as this no doubt includes several false positive associations. I recommend removing that comment in lieu of a statement of availability of the full association results.

This statement has now been removed and only results meeting $p < 5 \times 10^{-8}$ are included. Given there are no equivalent datasets from African populations, summary statistics derived from this study are a useful resource for others to use and thus we confirm that summary statistics from the full analyses will be made publicly available through the EBI GWAS catalog and is mentioned in our data availability statement (lines 585-587).

8. The number of previously reported KS variants the authors were able to assess seems low (8/31) given the density of genotyping and imputation in this sample. Can the authors provide an explanation as to why these variants were not detected and was any attempt made to test proxies for the missing variants?

We have now been able to impute classical HLA alleles and thus have increased the number of variants detected from 8 to 21 see lines (241-243). The remaining missing variants either did not meet out QC threshold or are in the KIR locus which we did could not reliably impute.

9. For completeness, Manhattan plots for the anti-EBV antibody GWAS should be included either in the main text or the supplement. Also, there were presumably no non-HLA associations detected in the is analysis? If so I think this should be stated.

Manhattan plots for EBV antibody GWAS are already included in the supplemental file 2 see supplementary Figure S5. No non-HLA associations were detected, statement added see lines 301-302.

Minor points:

In the introduction and discussion, the authors write: “We and others have reported associations in the HLA class II region with EBV EBNA-1 antibody response” but only provide a single reference. Reference to the work by the “and others” would be appropriate here.

This has now been adjusted to include other references. See references 11-14.

Given the results first organization of the manuscript, some methodological details should be included in each section. For example GPC should be defined with a brief description provided in the first results section to improve readability. Also, brief descriptions of the statistical methods used (single vs multiple regression, covariates included etc) would be helpful.

We have now adjusted the results section to include brief methodological details to improve clarity. See results lines 95-103: “ To investigate seroprevalence of infections, we tested serum samples from 4,365 individuals in the General Population Cohort (GPC), collected during medical survey round 22 (in 2011). GPC is a population-based cohort in Kyamulibwa, rural south-west Uganda, consisting of 25 neighbouring villages [26]. Participants were over the age of 13 years and mainly (>70%) belonged to the Baganda ethnolinguistic group. Villages were distributed across urbanicity quartiles reflecting shared ‘urban’ characteristics based on differences in economic activity, civil infrastructure, and availability of educational and healthcare services as previously described[27], with 28% of living in quartile 1(very rural i.e. no educational facilities and no households with electricity) (Table 1).”

And lines 135-137:” We selected age, sex, socio-demographic and infection status factors (Table 1) that could potentially influence serostatus or the levels of seroreactivity to KSHV antigens. Following a multivariate logistic regression analysis...”

Paul J McLaren, PhD

Reviewer #3 (Remarks to the Author):

Sallah et al. studied the factors associated with humoral immune response to infection by γ -herpesviruses in a large African cohort. The EBV analyses confirmed and expanded the known associations in the HLA class II region, while the KSHV analyses did not reveal any significant association after correction for multiple testing. A large African-based GWAS of the immune response to prevalent viral diseases is welcome. However, the paper suffers from a number of weaknesses.

Major comments:

1. Correlation and causation are often confounded is the way the results are presented. This study mostly shows associations and the wording should reflect this reality. For example, in the first paragraph of the discussion (line 282), “..., where we found HIV infection in particular influencing IgG antibody response levels to KSHV and EBV infection. It is interesting that while HIV infection lowered antibody responses to KSHV, the inverse was observed for EBV antibodies.”: HIV infection is highly

unlikely to influence antibody responses against EBV or KSHV in such a manner, but is rather an indirect epidemiological marker.

Although in very advanced HIV disease antibody levels can diminish (for example, we have seen KSHV seronegative KS patients, who are by definition infected, at very low CD4+ cell counts), an epidemiological association between HIV infection and antibody levels is possible as observed in Labo, et al. Mutual detection of Kaposi's sarcoma-associated herpesvirus and Epstein-Barr virus in blood and saliva of Cameroonians with and without Kaposi's sarcoma. *Int J Cancer*. 2019. Epub 2019/07/03. doi: 10.1002/ijc.32546. PubMed PMID: 31265124

We have rephrased the main text (lines 326-328) to avoid confusion: "This could be due to differential effect of shared epidemiological risk factors for HIV, EBV and KSHV, and/or as a result of the mutual inhibition of KSHV and EBV that has been observed in other studies"

2. Correction for multiple testing should be stringent and only significant results should be discussed at length. In the Results, the authors rightly state: "Correcting for multiple testing and accounting for the lower linkage disequilibrium (LD) in African populations the genome-wide significance threshold was adjusted to $p < 5 \times 10^{-9}$ ". However, they immediately follow by describing multiple non-significant associations. These "not-quite-significant" associations are presented again in the discussion. This is not only confusing to the reader, it also gives the impression that the authors don't really know what should be done about these results. Is there any reason to believe that the reported polymorphisms are worth investigating further, in which case they should indeed be mentioned in the main text of the paper? If not, they should only be available as part of the summary statistics of the study, without further emphasis.

We have removed SNPs $p < 1 \times 10^{-6}$ from the supplementary to avoid confusion (they can be found in the full association summary statistics). We now only include SNPs meeting the standard threshold of $p < 5 \times 10^{-8}$ as this is widely used in the field and the variants detected at this threshold fall within highly plausible regions which have strong priors such as HLA region. We believe these are strong candidates to prioritise for replication testing in other populations so we think that they are worth highlighting in the main text.

3. Results line 209: "In this cohort, 72% and 94% of individuals were categorized as KSHV and EBV seropositive". Why only 72% of KSHV seropositivity now? It is >90% in the first paragraph of the results.

Thank you for pointing out this discrepancy. This is an error and has been corrected in the text, see statement in results (lines 103-105): "In this study, 91% of individuals were categorized as seropositive for EBV based on based on detectable IgG levels against either -EBNA-1 or VCA and 91% categorised as seropositive for KSHV based on detectable IgG levels against either ORF73, K10.5 or K8.1 over pre-established cut-offs" and in discussion (lines 312-313) : "In the GPC, prevalence estimates of KSHV and EBV are high (>90%) and consistent with previous findings in Uganda"

4. Methods: it is not clear which method was used for which genome-wide analysis. There are two contradictory statements in the "genome-wide analyses" section: "We conducted analyses for all quantitative antibody traits for KSHV (N=4,365) and EBV (N=3,289) applying a linear regression model adjusting for age, age2 and sex in R" and "To control for cryptic relatedness and population

structure within the GPC, the GWAS was performed using the standard mixed model approach in GEMMA”.

Apologies for the lack of clarity. Prior to GWAS we first applied a linear regression adjusting for significant covariates. We then transformed the residuals for GWAS for which we used GEMMA for association as is standard. This statement has been rephrased in the methods to reflect both steps in that order.

See statement in methods (lines 512-519) “We conducted analyses for all quantitative antibody traits for KSHV(N=4365) and EBV (N=3,289 and N=4365 (including overlapping samples from pilot study)) by first applying a multivariable linear regression model adjusting for significant covariates in R. Residuals of MFI values used for analyses were then transformed using inverse, rank-based normalization in R to ensure a standard normal distribution for the phenotypes and confirmed by visualisation. To control for cryptic relatedness and population structure within the GPC, the GWAS of transformed phenotypes was performed using the standard mixed model approach in GEMMA”.

5. Multiple linguistic imprecisions and grammatical errors make some sections of the manuscript hard to follow. This should be corrected. Some examples:

We have now rephrased as appropriate in the text. See below

- Intro lines 44-45: “KSHV displays striking geographic variation that parallels KS incidence of disease caused by the virus”. Please rephrase

Lines 53-54: “KSHV displays striking geographic variation that parallels associated KS incidence”

- Intro lines 53 to 55: “For both viruses, B-lymphocytes act as a reservoir of latent infection and have been found to promote latency by subverting the host immune response and inhibiting lytic reactivation of each other in dually infected cells”.

Lines 60-61: “B-lymphocytes act as a reservoir of latent infection, in dually infected cells, both viruses subvert the host immune response and inhibit lytic reactivation of each other”

- Discussion line 311: “Here, we used IgG response traits as an intermediate phenotype for KS, seeing as sample sizes are limiting to conduct well powered GWASs for KSHV-associated diseases such as KS, and that previous studies have shown correlation of IgG levels with the development of KS.”

Lines 357-361: “Conducting GWASs for KSHV associated diseases is challenging as sample sizes are limiting to achieve power for reliable detection of loci. Prior to this study, GWAS had not been performed for any KSHV phenotype. As previous studies have shown correlation of IgG levels with the development of disease[55, 56], here we used IgG antibody responses as a proxy for disease progression”

Minor comments:

1. Abstract: “study” is missing in this sentence: “In 4,365 individuals from an African population cohort, we performed epidemiological analysis and genome-wide association (GWAS)...”

Thanks for pointing this out. We have amended.

2. Results: the global serology prevalence results are presented repeatedly, i.e. in Table 1, Figure 1 and in the text. This should be streamlined.

We have now removed figure 1A to avoid duplication and moved Figure1B to the supplementary.

3. Still in the Results: the long paragraphs “Environmental determinants of IgG serostatus to KSHV and EBV infections” and “Environmental determinants of IgG response levels to KSHV and EBV infections” are hard to read and are mostly descriptions of Figure 2 and Figure 3, respectively. They should be shortened to only highlight essential results. Figure 2 should also be redrawn to make it more informative (most ORs are very close to 1, so that part of the figure should be larger).

Thank you for the suggestions. We have incorporated your and other reviewers’ suggestions and redrawn Figures 2 and 3 to focus on only significant association and shortened the text to focus on essential findings.

4. Methods, line 426: “Of the merged datasets, 343 samples had overlapping genotype and sequence variant calls for which a final concordance of 93.1% was achieved for all SNPs.” This concordance rate sounds quite low. Please comment.

Apologies for the error, after removal of low frequency variants (maf<0.5%) the overall concordance of our final dataset was 98% we have now adjusted this in the methods (line 505). As we described previously in Gurdasani et al,2019, reference below) a minimum concordance threshold of 90% was required to abolish systematic effects observed between genotype array and sequence data on PCA. Following exclusion of all variants that showed < 90% concordance in genotypes between the sequence and imputed genotype data, PCAs did not show any systematic differences between imputed genotype and sequence data.

Reference:

Gurdasani et al,2019, Uganda Genome Resource Enables Insights into Population History and Genomic Discovery in Africa. Cell. 2019;179(4):984-1002 e36. Epub 2019/11/02. doi: 10.1016/j.cell.2019.10.004. PubMed PMID: 31675503.

REVIEWERS' COMMENTS:

Reviewer #1 (Remarks to the Author):

The authors have done a very good job addressing the comments and the manuscript reads clearly.

Reviewer #2 (Remarks to the Author):

I thank the authors for their careful attention to the points I raised. I am satisfied with all responses and have no further issues.

REVIEWERS' COMMENTS:

Reviewer #1 (Remarks to the Author):

The authors have done a very good job addressing the comments and the manuscript reads clearly.

We thank the reviewer for thoroughly reviewing our manuscript and their contributions to improving our manuscript.

Reviewer #2 (Remarks to the Author):

I thank the authors for their careful attention to the points I raised. I am satisfied with all responses and have no further issues.

We thank the reviewer for thoroughly reviewing our manuscript and their contributions to improving our manuscript.